# SPARSE-PIVOT: Dynamic correlation clustering for node insertions

Mina Dalirrooyfard [* 1]   Konstantin Makarychev [* 2]   Slobodan Mitrović [* 3]

## Abstract

We present a new Correlation Clustering algorithm for a dynamic setting where nodes are added one at a time. In this model, proposed by Cohen-Addad, Lattanzi, Maggiori, and Parotsidis (ICML 2024), the algorithm uses database queries to access the input graph and updates the clustering as each new node is added. Our algorithm has the amortized update time of $O_\varepsilon(\log^{O(1)}(n))$. Its approximation factor is $20 + \varepsilon$, which is a substantial improvement over the approximation factor of the algorithm by Cohen-Addad et al. We complement our theoretical findings by empirically evaluating the approximation guarantee of our algorithm. The results show that it outperforms the algorithm by Cohen-Addad et al. in practice.

## 1. Introduction

In this paper, we present a new dynamic algorithm with node updates for the Correlation Clustering problem with complete information.[1] Correlation Clustering is a well-studied problem that seeks to partition a set of objects into clusters based on their similarity. The problem is defined on a set of items represented as nodes in a graph, with similarity information provided through a set of edges. We assume that all pairs of nodes are classified as either "similar" or "dissimilar" by a noisy classifier. For every pair of similar nodes $u$ and $v$, there is an edge $(u, v)$ between them (sometimes referred to as a positive edge). For every pair of dissimilar nodes $u$ and $v$, no edge exists between them (such pairs are sometimes called negative edges). The objective is to find a clustering that minimizes the number of disagreements

*Equal contribution   [1]Machine Learning Research, Morgan Stanley, Canada.   minad@mit.edu. [2]Northwestern University, Chicago, USA. konstantin@northwestern.edu.   [3]University of California, Davis, USA. smitrovic@ucdavis.edu. Correspondence to: Mina Dalirrooyfard <minad@mit.edu>.

*Proceedings of the 42$^{nd}$ International Conference on Machine Learning*, Vancouver, Canada. PMLR 267, 2025. Copyright 2025 by the author(s).

[1]By complete information, it is meant that for each pair of nodes there is information on their similarity.

with the given edge set. An edge $(u, v)$ disagrees with the clustering if $u$ and $v$ are placed in different clusters, while a non-edge $(u, v)$ disagrees if $u$ and $v$ are placed in the same cluster.

The problem was introduced by Bansal, Blum, and Chawla (2004). It can be easily solved if a disagreement-free clustering exists, as each cluster then corresponds to a connected component of the similarity graph $G$. However, when the classifier makes mistakes, and a disagreement-free solution does not exist, the problem becomes NP-hard. In their original paper, Bansal, Blum, and Chawla proposed a constant-factor approximation, which was subsequently improved in a series of works Charikar et al. (2005); Demaine et al. (2006); Chawla et al. (2015); Cohen-Addad et al. (2022; 2023); Cao et al. (2024). In 2005, Ailon, Charikar, and Newman introduced combinatorial and LP-based algorithms with approximation factors of 3 and 2.5, respectively. The best-known approximation factor for Correlation Clustering with Complete Information is currently 1.437 (Cao, Cohen-Addad, Lee, Li, Newman, and Vogl, 2024). Nevertheless, the combinatorial algorithm by Ailon et al. (2008), known as PIVOT, remains one of the preferred choices in practice due to its simplicity and good empirical performance.

Researchers have proposed various variants of PIVOT that operate in parallel and streaming settings (Bonchi et al., 2014; Chierichetti et al., 2014; Pan et al., 2015; Cohen-Addad et al., 2021; Cambus et al., 2022; Behnezhad et al., 2022; 2023; Cambus et al., 2022; Chakrabarty & Makarychev, 2023). Recently, there has been growing interest in algorithms that support dynamic updates. Consider a scenario where similarity information is received over time, requiring the clustering to be updated dynamically. Dalirrooyfard, Makarychev, and Mitrovic (2024) showed how to maintain a $(3 + \varepsilon)$-approximation clustering with constant update time per edge insertion or deletion (see also papers by Behnezhad et al. (2019) and Chechik & Zhang (2019)).

Cohen-Addad, Lattanzi, Maggiori, and Parotsidis proposed an algorithm designed for a setting where nodes and edges are stored in a database. Over time, new nodes are added to the database along with their incident edges. After nodes are added, the dynamic algorithm updates the existing clustering. In this model, each edge is included in the database immediately after both its endpoints are inserted; no ad-

ditional edges can be inserted or deleted afterward. The sequence of inserted nodes is non-adaptive, meaning it does not depend on the decisions of the clustering algorithm. Formally, we assume that the graph and the order of node arrivals are fixed in advance. Cohen-Addad et al. showed how to achieve a constant (albeit very large) approximation with an update time of $\log^{O(1)} n$, measured in terms of database operations defined as follows: (1) retrieving the degree of a node $v$; (2) selecting a random neighbor of $v$; and (3) checking whether two nodes $u$ and $v$ are connected by an edge. The database model was introduced by Assadi and Wang (2022). The algorithm by Cohen-Addad et al. was the first dynamic algorithm for node insertions with sublinear update time. Although it provides a constant approximation, the proof suggests that the constant is very large (the paper does not estimate it). In this work, we propose a $(20 + \varepsilon)$-approximation algorithm for Correlation Clustering with an update time of $O_\varepsilon(\log^{O(1)} n)$ database operations per node insertion.

In addition to node insertion, the algorithm by Cohen-Addad et al. supports deleting random nodes. Our algorithm supports a slightly weaker type of deletion: *soft deletions* of random vertices. When a node is soft-deleted from the graph, it initially remains in the database but is marked as soft-deleted. Moreover, the classifier continues creating edges between newly arriving and soft-deleted nodes. Only when the algorithm requests their deletion are they purged from the database.

To motivate the model, consider the following example: An online store adds new items to its stock daily and aims to cluster all items based on similarity. Whenever a new item is added, the store runs a classifier to identify items similar to the new one. A record for the new item is then created and inserted into the database, along with edges connecting it to similar items. Instead of reclustering the entire dataset, the dynamic algorithm efficiently updates the clustering, requiring only $O(\log^{O(1)} n)$ operations per item insertion.

## 2. Algorithm

Our algorithm is based on the 5-approximation variant of PIVOT, developed by Behnezhad, Charikar, Ma, and Tan (2023) for the semi-streaming model. Their algorithm in the static setting works as follows: First, it selects a random ordering $\pi$ of all nodes. For each node $u$, it picks the neighbor of $u$ with the smallest rank and stores this neighbor in the pivot array $p$: $p(u) = \arg\min_{w \in N(u)} \pi(w)$. Next, the algorithm identifies all nodes $u$ such that $p(u) = u$, which we refer to as pivots. We call $p(v)$ the pivot for node $v$. Then, the algorithm creates a new cluster for every pivot $u$ and assigns to it nodes $v$ with $p(v) = u$. All remaining unassigned nodes are placed in singleton clusters. Note that for each node $u$ in a singleton cluster, $p(u)$ remains equal

to the neighbor of $u$ with the smallest rank. To reiterate, in this variant of PIVOT, each vertex $u$ belongs to the cluster of $p(u)$ if $p(u) = p(p(u))$, and to a singleton cluster if $p(u) \neq p(p(u))$.

Behnezhad, Charikar, Ma, and Tan (2023) provide a dynamic edge-insertion version of this algorithm with constant update time. A dynamic implementation of this algorithm for node insertions can be easily inferred and is presented in Algorithm 1. In this version, when the node $u$ being inserted is a pivot, the algorithm runs the EXPLORE process. This process updates the pivot of each neighbor $w$ of $u$ and reassigns $w$ and its neighbors to the cluster of $u$ or singleton clusters, if necessary. We refer to the clustering produced by this algorithm as REFERENCE CLUSTERING, as our algorithm aims to approximate this clustering.

---

**Algorithm 1** Insertion for REFERENCE CLUSTERING

1: **input** node $u$, graph $G$, ordering $\pi$, pivot array $p$.
2: Compute $p(u) = \arg\min_{w \in N[u]} \pi(w)$.
3: **if** $p(u) = u$:
4:     Mark $u$ as a pivot.
5:     Create a new cluster with $u$ in it.
6:     Run EXPLORE($u, G, \pi, p$)
7: **else if** $p(u)$ is a pivot:
8:     Put $u$ in $p(u)$'s cluster.
9: **else**:
10:     Make $u$ a singleton.

---

### 2.1. Making the REFERENCE CLUSTERING algorithm faster

The challenge with REFERENCE CLUSTERING is that, after each insertion, it scans the entire neighborhood of the inserted node $u$, which can be as large as $\Theta(n)$, where $n$ is the size of the current graph. This makes this REFERENCE CLUSTERING prohibitively expensive. To improve the efficiency, we perform this exhaustive search only for pivots. For non-pivot nodes $u$, we attempt to recover the unknown pivot $v$ by sampling $O(\log n)$ neighbors of $u$, examining the set of pivots of these sampled neighbors, and setting $p(u)$ to the neighbor with the smallest ranked among the neighbors scanned in this process (which could be the samples or their

---

**Algorithm 2** EXPLORE

1: **input** pivot $u$, graph $G$, ordering $\pi$, pivot array $p$.
2: **for all** $w \in N[u]$:
3:     **if** $\pi(p(w)) > \pi(u)$:
4:         **if** $w$ is a pivot, i.e., $p(w) = w$:
5:             **for all** $z$ where $p(z) = w$:
6:                 **if** $z \in N(u)$, then $p(z) \leftarrow u$,
7:             **else** make $z$ a singleton
8:     $p(w) \leftarrow u$.

pivots). We show that this approach succeeds when the cluster $C$ in the reference clustering is sufficiently dense and $u$ does not have too many neighbors outside of $C$.

Instead of selecting a random ordering of vertices $\pi$, our algorithm assigns each value $\pi(u)$ uniformly at random from the interval $[0, 1]$. This simplifies maintaining a random ordering in dynamic settings, where the exact number of nodes is not known in advance and also makes the logic of our algorithm a bit simpler.

We present our algorithm for node insertion in Algorithm 3. We use three main ideas:

(1) For vertices $u$ with $\pi(u) \leq \frac{L}{d(u)}$, where $L = O(\log n)$, we run EXPLORE on $u$ if $u$ is a pivot, similarly to REFERENCE CLUSTERING. If $u$ is not a pivot, we run EXPLORE on its pivot to update its cluster.

(2) For vertices $u$ with $\pi(u) > \frac{L}{d(u)}$, we find a pivot by examining the set of pivots of $\Theta(\log n)$ random neighbors of $u$ and selecting the neighbor of $u$ with smallest ranked in that set as a pivot for $u$.

(3) In the obtained tentative clustering, we identify certain nodes, remove them from their clusters, and place them into singleton clusters. Specifically, we remove nodes whose in-cluster degree is below a threshold $t$. The optimal value of $t$ is determined by trying out $O(\log n)$ possible choices. In particular, when a new cluster is made upon arrival of a pivot node, the subroutine BREAK-CLUSTER removes some nodes from the cluster and makes them singletons. When the node inserted is not a pivot and is assigned to pivot $v$, the subroutine UPDATE-CLUSTER updates the set of nodes assigned to $v$ that are put in singletons upon insertion of the new node (See Appendix A for details).

We now provide some motivation for the algorithm. First, we observe that the expected time required for finding pivots in items (1) and (2) is $\log^{O(1)} n$. In item (1), we spend $d(u) \log n$ time with probability $\log n / d(u)$. In item (2), we always spend $O(\log n)$ time by examining $O(\log n)$ random neighbors and their pivots.

We run the step described in item (3) after (roughly) every $\epsilon s$ insertions, where $s$ is the size of the cluster. In this step, we test $O(\log n)$ different thresholds $t$ and, for each choice of $t$, estimate the cost of the split defined by $t$ by sampling random edges within the cluster (see Appendix A for details). Hence, the total running time of one such step is $s \log^{O(1)} n$, and the amortized cost per insertion is $\log^{O(1)} n$.

We now discuss the approximation factor. Item (1) ensures that all pivots in the reference clustering are marked as pivots by our algorithm with high probability. Specifically, for every pivot $u$ in the reference clustering, we have $\pi(u) \leq L/d(u)$ with probability at least $1 - \text{poly}(1/n)$, ensuring

that EXPLORE is called on every pivot $u$. This follows from the observation that if $\pi(u) > L/d(u)$, then $\pi(v) > L/d(u)$ for all neighbors $v \in N[u]$. The probability of this event is at most $(1 - L/d(u))^{d(u)} \leq \text{poly}(1/n)$.

We define *good* nodes as follows: loosely speaking, a node is good if it is connected to most nodes within its cluster and to relatively few outside of it. We show that *good* nodes in our clustering are assigned the same pivots as in the reference clustering (when our algorithm and REFERENCE CLUSTERING use the same ordering $\pi$). For now, let us assume that all nodes are good. If a good node $u$ arrives after its pivot $w$, then $u$ will join $w$'s cluster when $w$ scans all its neighbors, including $u$. If $u$ arrives after $w$ while more than 25% nodes in $w$'s cluster have yet to arrive, then for at least one node $v$ arriving after $u$, we will call EXPLORE, which will assign $u$ the correct pivot $w$. Here, we rely on the assumption that all nodes are good. Finally, if $u$ arrives among the last 25% of nodes in $w$'s cluster, then by the time $u$ arrives, the algorithm will have assigned the correct pivot $w$ to most of its neighbors in the reference cluster. Consequently, a high fraction of $u$'s neighbors will have $w$ as their pivot, and some of these neighbors will be included in the random sample of neighbors (see item (2)). Hence, $u$ will also be assigned the correct pivot. Our full analysis can be found in Section 3.

Let us now consider bad (not good) nodes. These nodes incur a very substantial cost in the reference clustering. If we could remove them from their reference clusters and place them into singleton clusters, the overall clustering cost would not increase significantly—in fact, it might even decrease. Unfortunately, our algorithm cannot identify these bad nodes; as a result, they may join clusters other than their own reference clusters. This can substantially increase the cost of those clusters. To address this issue, we partition each tentative cluster into two parts (see item (3)): the first part remains a cluster, while the second part is broken into singleton clusters.

Theorem 2.1 provides the approximation guarantees of our algorithm which we call SPARSE-PIVOT.

**Theorem 2.1.** *For any $\epsilon < 1/1000$, the expected cost of* SPARSE-PIVOT *with parameter $\epsilon$ is at most $4(1 + O(\epsilon))$ times the expected cost of the* REFERENCE CLUSTERING.

Our algorithm for soft deletions is very simple: We ignore them! In fact, we recompute the whole clustering again after $\Theta(\varepsilon)N$ many updates, where $N$ is the number of nodes when we last recomputed the clustering. The recomputation is only necessary for deletions. Our running time guarantees are provided in Theorem 2.2.

**Theorem 2.2.** *Let $T$ be the total number of updates. With high probability, Algorithm 3 runs in amortized* $\text{poly}(\log T, \frac{1}{\varepsilon})$ *time.*

**Algorithm 3** INSERT-NODE-SPARSE-PIVOT

1: **input** node $u$ to be inserted, current graph $G$ of size $n$, ordering $\pi$, pivot vector $p$.
2:     Let $\pi(u) \in [0,1]$ be chosen uniformly at random.
3:     Let $p(\cdot)$ indicate the lowest rank neighbor of each node.
4:     Let $B_v$ indicate the set of nodes with pivot $v$, and let $C_v \subseteq B_v$ be the nodes of $B_v$ that are in $v$'s cluster.
5:     **if** $\pi(u) \leq \frac{L}{d(u)}$:
6:         Find $v = \arg\min_{w \in N[u]} \pi(w)$.
7:         **if** $v = u$:
8:            Make $u$ a pivot: $p(u) \leftarrow u$, and make a new cluster with $u$ in it: $B_u = \{u\}$, $t_v = 0$
9:            EXPLORE$(u, G, \pi, p)$
10:            $C_u \leftarrow$ BREAK-CLUSTER$(B_u)$.
11:         **else if** $v \neq u$ and $v$ is a pivot:
12:            $p(u) \leftarrow v$, $B_v \leftarrow B_v \cup \{u\}$.
13:            $C_v \leftarrow$UPDATE-CLUSTER$(u, B_v)$
14:            **if** $d(v) \leq \frac{L}{\pi(u)}$:
15:                EXPLORE$(v, G, \pi, p)$
16:         **else if** $v \neq u$ and $v$ is not a pivot:
17:            make $u$ a singleton.
18:     **else**:
19:         Let $S$ be a $O(\log n)$-sized sample of $N[u]$.
20:         Let $s^* = \arg\min_{s \in S, \{p(s), u\} \in E(G)} \pi(p(s))$. Let $v := p(s^*)$
21:         **if** $\pi(v) < \pi(u)$:
22:            $p(u) \leftarrow v$, $B_v \leftarrow B_v \cup \{u\}$.
23:            $C_v \leftarrow$UPDATE-CLUSTER$(u, B_v)$
24:         **if** $u$ did not get clustered:
25:            make $u$ a singleton, $u$ does not have a pivot.

## 2.2. Analysis Preliminaries

We use subscript $ref$ to refer to REFERENCE CLUSTERING. We fix time, and we compare the cost of our algorithm to the cost of REFERENCE CLUSTERING at this time. We refer to the current graph as $G$. For a node $v$, ordering $\pi$ and clustering algorithm $A$, let $C_A^\pi(v)$ be the cluster of $v$ in $A$ with respect to the ordering $\pi$. We drop the superscript $\pi$ and subscript $A$ when it is clear from the context. The clustering algorithms that we consider throughout our analysis all define pivots for all non-singleton clusters.

Given a ordering $\pi$ and clustering algorithm $A$, let $p_A^\pi(u)$ denote the pivot of $u$ chosen by algorithm $A$. For a set of nodes $S$, let $d_S(v)$ be the degree of $v$ in $S$, and let $d(v)$ be the degree of $v$ in the graph $G$. We will classify nodes depending on how their neighborhoods intersect the cluster to which they are assigned.

**Definition 2.3.** Let $\alpha < 1$ and $\beta > 1$. Let $A$ be a clustering algorithm. For a fixed ordering $\pi$, we call vertex $u$

- *A-light* if $d_C(u) \leq \frac{|C|}{3}$, where $C = C_A^\pi(u)$.

- $(A, \alpha)$-*poor* if $d(u) \leq \alpha d(p_A(u))$ and $u$ is not light.

- *A-heavy* if $d(u) \geq \beta|C|$, where $C = C_A^\pi(u)$.

- *A-bad* if $u$ is $(A, 3\alpha\beta)$-poor, *A*-heavy or *A*-light, and *A-good* otherwise.

- *A-lost* if the number of $A$-bad neighbors of $u$ in $C_A^\pi(u)$ is at least $\beta$ times the number of $A$-good neighbors of $u$ in $C_A^\pi(u)$.

We drop $A$- if the clustering algorithm is clear from the context.

**Definition 2.4** (Poor clusters). For a fixed ordering $\pi$ and clustering algorithm $A$, let $C$ be a cluster with pivot $v$. We call $C$ an $(A, \alpha)$-poor cluster if $C$ has at least one $(A, \alpha)$-poor node.

**Definition 2.5** (Good and bad clusters). Let $\gamma < 1$ be a constant. For a fixed ordering $\pi$ and clustering algorithm $A$, we call a cluster $C_A^\pi$ good if it is not $(A, \alpha)$-poor, and it has at least $\gamma|C_A^\pi|$ good nodes. Otherwise, we call it bad.

*Remark* 2.6. Note that instead of fixing the clustering algorithm $A$ and ordering $\pi$, we can still have the above definitions if we fix the clusters and pivots.

**Definition 2.7** (Cost of a cluster). Given a clustering, the cost of a cluster $C$ is the number of non-edges inside $C$, and half of the number of edges with exactly one endpoint in $C$.

The cost of a clustering is the sum of the cost of its clusters.

## 3. Analysis

### 3.1. Analysis outline

First, we explain the intuition behind classifying nodes in Definition 2.3. Let the cost of a node be half the number of non-neighbors it has in its cluster, plus half the number of neighbors it has outside the cluster. Note that the sum of the costs of nodes in a cluster equals the cost of the cluster. Consider $A$ to be the reference clustering in Definition 2.3. A light node in cluster $C$ has a lot of non-edges attached to it in $C$, and a heavy node has a lot of edges attached to it that leave the cluster $C$. So both have a high cost. In fact, we show that if we make them singletons, the cost of the clustering does not change much. So, in a sense, in our algorithm, we do not care how $ref$-heavy or $ref$-light nodes are being clustered as long as their cost is somewhat comparable to their cost as singletons.

Consider poor nodes. A poor node has a much lower degree than its pivot. We show that if a cluster has at least one poor node, then any node in this cluster that is not heavy or light must be poor (Lemma 3.1). Then we show that, in fact, if we

make the whole cluster singleton, the cost of the clustering does not change much *on average*. This is because the pivot of this cluster has a very high cost, and any node becomes the pivot of a poor cluster with low probability.

In summary, we have shown that if we make the bad nodes (heavy, poor, or light) in the reference clustering singleton, the cost of the clustering does not change much (Lemma 3.2). We further show that we can make all the nodes in a bad cluster singleton as well since the cost of this cluster is already too high. We call this clustering $ref'$.

We then show that in SPARSE-PIVOT, not only do we detect pivots correctly with high probability (Lemma 3.4), but also all the good nodes that are not lost, i.e., they do not have many bad neighbors, are assigned the correct pivot. We show this in two parts: Consider a pivot $v$, and suppose $C$ is the set of all the $ref$-good nodes with pivot $v$ that are not $ref$-lost. First, we show that if a good node $u \in C$ arrives rather early compared to other nodes in $C$, at some point its pivot runs EXPLORE and it detects if $u$ is clustered incorrectly (Lemma 3.5). If $u$ arrives rather late, then the sampling procedure will hit one of the neighbors of $u$ in $C$ that is correctly clustered and so correctly assigns $v$ as the pivot of $u$ (Lemma 3.6).

Now since we cluster the $ref$-good nodes correctly with high probability, if we could detect $ref$-bad nodes in SPARSE-PIVOT, we could make them singletons, and thus get $ref'$. However, instead, if $B_v$ is the set of nodes that have $v$ as their pivot, we make a dense subset $C_v$ of $B_v$ one cluster, and make the rest of $B_v$ singleton. This step is crucial since there might be many (bad or lost) nodes incorrectly assigned to $v$ that increase the number of the non-edges in $B_v$ significantly. We show that the cost of making $B_v \setminus C_v$ singleton is at most 4 times the cost of making the $ref$-bad nodes in $B_v$ singleton (Lemma 3.7).

Now we begin our formal analysis. Let $\beta \geq \frac{4+\epsilon}{\epsilon}$, $\alpha < \min(\frac{\epsilon}{24\beta}, \frac{1}{39\beta})$, and $\gamma \leq \frac{\epsilon}{2}$. Let $L \geq \frac{4c\beta}{\epsilon\gamma\alpha}\log n$ for some arbitrary large constant $c$. Let the number of samples when the inserted $u$ satisfies $\pi(u) > L/d(u)$ be at least $100\log(\frac{1}{1-x}) = O(\log n)$, where $x = (\frac{1}{\beta+1} - \epsilon)/\beta$.

### 3.2. Making all the bad and lost nodes singleton in Reference clustering

We show that if a cluster has one $\alpha$-poor node, then all the nodes in that cluster are $3\alpha\beta$-poor.

**Lemma 3.1.** *Let $\pi$ be an ordering and $A$ be a clustering algorithm. If $C_A^\pi$ is an $(A, \alpha)$-poor cluster, then any $u \in C_A^\pi$ which is not light or heavy is $(A, 3\alpha\beta)$-poor.*

*Proof.* Let $v$ be the pivot of $C = C_A^\pi$. First, since $C$ is an $(A, \alpha)$-poor cluster there is a node $w \in C$ that is $(A, \alpha)$-poor. Since, by definition of poor nodes, $w$ is not light,

we have that $|C|/3 \leq d(u) \leq \alpha d(v)$. So $3\alpha d(v) \geq |C|$. Now for any $u$ that is not heavy we have $d(u) \leq \beta|C| \leq 3\alpha\beta d(v)$. Thus if $u$ is not light, then $u$ is $(A, 3\alpha\beta)$-poor. $\square$

The following lemma shows that if, in REFERENCE CLUSTERING, we make all the bad nodes and lost nodes singleton, the cost only increases by a factor of $(1 + O(\epsilon))$.

**Lemma 3.2** (Cost of making bad and lost nodes singletons). *Consider a clustering algorithm $A$ where the probability of any node $v$ being a pivot is at most $1/d(v)$. Let $B$ be the algorithm that first runs $A$, and then makes $A$-heavy, $A$-light, $(A, 3\alpha\beta)$-poor and $A$-lost nodes singletons. Then $\mathbf{E}_\pi[cost_B] \leq (1 + 7\epsilon)\mathbf{E}_\pi[cost_A]$*

*Proof.* Let $\alpha' = 3\alpha\beta$, and let $A'$ be the algorithm that runs $A$ and then makes the $A$-heavy and $A$-light nodes singleton. Let $A''$ be the algorithm that runs $A'$, and makes all the $(A, \alpha')$-poor nodes singleton. For the clarity and brevity of notation, we use $\delta \overset{\text{def}}{=} 4\alpha'\frac{1+3/2\cdot\alpha'}{1-5/2\cdot\alpha'}$. For any fixed ordering $\pi$, by Lemma C.2, $cost_{A'}^\pi \leq \frac{\beta+1}{\beta-1}cost_A^\pi$, and

$$cost_{A'} \leq \frac{\beta+1}{\beta-1}cost_A.$$

Now note that the probability of any node $v$ being a pivot in $A'$ is at most the probability of any node being a pivot in $A$, which is at most $1/d(v)$. The cost of making $(A, \alpha')$-poor nodes singletons is, at most, the sum of their degrees. By Lemma C.1, that expected sum of $(A, \alpha')$-poor nodes degrees is upper bounded by $\delta \cdot \mathbf{E}_\pi[cost_A]$. Hence, we have

$$\mathbf{E}_\pi[cost_{A''}] \leq (1 + \delta)\mathbf{E}_\pi[cost_{A'}].$$

Finally, note that any $A$-lost node is a $A''$-light node or $A''$-heavy node with parameter $\beta/3$. To see this, fix a ordering $\pi$. Consider a $A$-lost node $u$, and let $C = C_A^\pi(u)$. Let $C' \subseteq C$ be the set of $A$-bad nodes in $C$. Let $C'' := C_{A''}^\pi(u) = C \setminus C'$. By the definition of $A$-lost nodes, we have $d_{C'}(u) \geq \beta d_{C''}(u)$. If $u$ is not $A''$-light, then $d_{C''}(u) \geq |C''|/3$. So $d(u) \geq d_{C'}(u) \geq \beta d_{C''}(u) \geq \frac{\beta}{3}|C''|$. So $u$ is $A''$-heavy node with parameter $\beta/3$. By Lemma C.2, $cost_B^\pi \leq \frac{\beta+3}{\beta-3}cost_{A''}^\pi$. So $cost_B \leq \frac{\beta+3}{\beta-3}cost_{A''}$. Putting all the steps together yields

$$\mathbf{E}_\pi[cost_B] \leq \frac{\beta+1}{\beta-1} \cdot (1+\delta) \cdot \frac{\beta+3}{\beta-3} \cdot \mathbf{E}_\pi[cost_A].$$

Now since $\beta \geq \frac{4+\epsilon}{\epsilon}$, we have $\frac{\beta+1}{\beta-1} < \frac{\beta+3}{\beta-3} \leq (1+\epsilon)$, and since $\alpha' < 1/13$, we have $\frac{1+3/2\cdot\alpha'}{1-5/2\cdot\alpha'} < 2$, and $\alpha' < \epsilon/8$ gives us $4\alpha'\frac{1+3/2\cdot\alpha'}{1-5/2\cdot\alpha'} < \epsilon$. So $\mathbf{E}_\pi[cost_B] \leq (1 + \epsilon)^3\mathbf{E}_\pi[cost_A] \leq (1 + 7\epsilon)\mathbf{E}_\pi[cost_A]$. $\square$

**Lemma 3.3.** *Consider a fixed ordering $\pi$ and a clustering algorithm $A$. If $B$ is a clustering algorithm that runs $A$ and makes all $A$-bad and $A$-lost nodes, as well as all the nodes in bad clusters in $A$ singleton, then $cost(B) \leq (1 + 8\epsilon)cost(A)$.*

*Proof.* Let $A'$ be the algorithm that runs $A$ and makes the $A$-bad nodes and $A$-lost nodes singleton. Then we can see $B$ as running $A'$ and making all the good nodes remaining in $A$-bad clusters singleton. Note that by Lemma 3.2 $cost(A') \leq (1 + 7\epsilon)cost(A)$. Note that in $A'$ all poor clusters are singletons, since a poor cluster has at least one $(A, \alpha)$ poor node, and by Lemma 3.1 all the non-pivot nodes in this poor cluster that are not heavy or light are $(A, 3\alpha\beta)$-poor. By definition of bad nodes, all the nodes in a poor cluster are bad.

Consider a bad cluster $C$ in $A$. The cost of $C$ in $A$ is at least $\frac{2}{3}(1 - \gamma)|C|^2$ by Lemma C.3. Making the good nodes in $C$ singleton in $A'$ adds at most $\gamma^2|C|^2$ to the cost of $A'$ since there are at most $\gamma|C|$ good nodes in $C$, and the only cost added by making these nodes singleton is through the edges between them. So the total cost added to the cost of $A'$ by making these good nodes singleton is at most $\frac{3\gamma^2}{2(1-\gamma)}cost(A)$. So $cost(B) \leq cost(A') + \frac{3\gamma^2}{2(1-\gamma)}cost(A)$. Since $\gamma \leq \epsilon/2$, we have $\frac{3\gamma^2}{2(1-\gamma)} \leq \frac{4\gamma^2}{(1-\gamma)^2} \leq \epsilon^2 < \epsilon$. So $cost(B) \leq (1 + 8\epsilon)cost(A)$. $\square$

### 3.3. SPARSE-PIVOT comparison to REFERENCE CLUSTERING

Let $P^\pi_{alg}$ be the pivot set of our algorithm and $P^\pi_{ref}$ be the pivot set of the reference clustering.

**Lemma 3.4.** *Let $L' = L/2\beta$. If $v \in P^\pi_{ref}$, then $\pi(v) \leq L'/d(v)$ holds with probability at least $1 - 1/n^{c-1}$.*

Note that not only does Lemma 3.4 say that each pivot in REFERENCE CLUSTERING is also a pivot in SPARSE-PIVOT, but it also provides a stronger guarantee on $\pi(v)$.

For the next Lemma, note that the definitions of light, poor, etc, are well-defined if the clustering is fixed (and not necessarily the ordering $\pi$). We show that with high probability, one of the good nodes in $C_{ref}(v)$ triggers EXPLORE function for $v$ so that $v$ can correct its cluster.

**Lemma 3.5.** *Fix a clustering and its pivots that REFERENCE CLUSTERING algorithm can produce. Consider a pivot $v$ whose cluster $C_{ref}(v)$ is good. Let $C_{ref}(v)[good]$ be the good nodes in $C_{ref}(v)$. Then, with high probability, our algorithm assigns $v$ as the pivot of $u$, for any $(1 - \epsilon)|C_{ref}(v)[good]|$ first nodes $u$ of $C_{ref}(v)[good]$. The "first" here is taken with respect to the dynamic ordering and the probability taken over rankings that produce the fixed clustering.*

Lemma 3.6 shows that SPARSE-PIVOT identifies the pivot of all the good nodes that are not lost correctly.

**Lemma 3.6.** *Fix a clustering and its pivots that the reference algorithm can produce. Let $v$ be a pivot in that reference clustering and $u$ be in the $v$'s cluster. With high probability, any $u$ that is ref-good and not ref-lost is assigned to $v$ by Algorithm 3.*

*Proof.* Let $C = C_{ref}(v)$. First, if $u$ is among the first $(1 - \varepsilon)$ fraction of the nodes of $C[good]$ with respect to the dynamic ordering, then by Lemma 3.5, with high probability Algorithm 3 assigns $u$ to $v$'s cluster.

So, second, consider the case when $u$ is in the last $\varepsilon$ fraction of $C[good]$. Let $D$ refer to the $ref$-good neighbors of $u$ that are among the first $1 - \varepsilon$ fraction of the nodes of $C[good]$. By Lemma 3.5, with high probability Algorithm 3 assigns the nodes in $D$ to $v$'s cluster. We now lower-bound $|D|$. We will use that lower bound to show that the sampling process in Algorithm 3 (when $\pi(u) < L/d(u)$) will sample at least one node from $D$, and hence assign $v$ to $u$'s cluster.

Since $u$ is not $ref$-lost, by Definition 2.3, $u$ has at most $\beta$ times more $ref$-bad than $ref$-good neighbors in $C$. Hence, at least $1/(1 + \beta) \cdot |C|$ neighbors of $u$ in $C$ are $ref$-good. Also, observe that $1/(1 + \beta) \cdot |C| - \varepsilon \cdot |C[good]| \geq (1/(1 + \beta) - \varepsilon) \cdot |C|$ of those neighbors are among the first $1 - \varepsilon$ fraction of $C[good]$. Therefore, $|D| \geq (1/(1 + \beta) - \varepsilon) \cdot |C|$. On the other hand, since $u$ is a $ref$-good node, we have that $d(u) < \beta \cdot |C|$.

Finally, we conclude that

$$\frac{|D|}{d_{insert}(u)} \geq \frac{|D|}{d(u)} \geq \frac{1/(1 + \beta) - \varepsilon}{\beta}.$$

where $d_{insert}(u)$ is the degree of $u$ at the time of insertion. Let $\bar{n}$ be the number of nodes when inserting $u$. If $x := \frac{1/(1+\beta) - \varepsilon}{\beta}$, the probability of not sampling any node in $D$ is at most $(1 - x)^{|S|}$, where $S$ is the sample set. Recall that $|S| \geq 100 \log(\frac{1}{1-x}) \cdot \log \bar{n}$, so this probability is at most $1/\bar{n}^{100}$. Note that if $n$ is the current number of nodes, $\bar{n} \geq n/(1 + \epsilon)$ since we recompute everything when the number of updates is at most $\varepsilon$ times the number of nodes at last RECOMPUTE. So the probability that $u$ is clustered correctly is at most $1 - (1 + \epsilon)/n^{100}$. $\square$

Next, Lemma 3.7 aids us to compare REFERENCE CLUSTERING with SPARSE-PIVOT clustering. Given pivot $v$ and set $B_v$, let $C_t$ be the set of nodes in $B_v$ with degree at least $t$. Let $cost(B_v|C_t)$ be the cost of the clustering on $B_v$ where all the nodes in $C_t$ are clustered as one cluster and the nodes in $B_v \setminus C_t$ as singletons. In particular, this cost equals half of the number of edges from $B_v$ to outside of $B_v$, plus the number of edges with at least one endpoint in $B_v \setminus C_t$, plus the number of non-edges in $C_t$.

**Lemma 3.7.** *Consider a pivot $v$, and let $C^*$ be the set of good nodes that are not lost in $C_{ref}(v)$. There is a threshold $t \in \{1, (1+\epsilon), \ldots, (1+\epsilon)^{\lceil \log n \rceil + 1}\}$, such that $cost(B_v|C_t) \leq \frac{4}{1-2\varepsilon} cost(B_v|C^*)$.*

### 3.4. Putting it all together

*Proof of Theorem 2.1.* We refer to REFERENCE CLUSTERING as $ref$ and the clustering of SPARSE-PIVOT by $B$. Let $A$ be the clustering algorithm that runs REFERENCE CLUSTERING to obtain $ref$, and then makes all $ref$-bad, $ref$-lost nodes as well as all the nodes in $ref$-bad clusters singleton. By Lemma 3.3, $\mathbb{E}(cost(A)) \leq (1+8\epsilon)\mathbb{E}(cost(ref))$. Note that all the nodes that are not singletons in $A$ are good nodes that are not lost and are not in bad clusters in $ref$.

Next we show that $\mathbb{E}(cost(B)) \leq 4(1 + O(\epsilon))\mathbb{E}(cost(A))$. Consider a pivot $v$ in REFERENCE CLUSTERING. By Lemma 3.4 $v$ is also a pivot in SPARSE-PIVOT with high probability. We will compare clustering costs by dividing up the clusters into groups: We consider all the nodes that are in $B_v$ for a pivot $v$ together and compare the cost of clustering these nodes in $A$ and in $B$. We have two cases:

**Case 1: $C_{ref}(v)$ is a good cluster.** Consider $B_v$, the set of nodes that are assigned to $v$ as their pivot, and $C_v \subseteq B_v$, the set of nodes in $B_v$ that are clustered with $v$ and are not singletons.

Let $C_t = \{u \in B_v | d(u) \geq t\}$, and let $cost(B_v|C_t)$ be the cost of clustering all nodes in $C_t$ as one cluster and the nodes in $B_v \setminus C_t$ as singletons. Let $t^*$ be the threshold in $\{1, (1+\epsilon), \ldots, (1+\epsilon)^{\lceil \log n \rceil + 1}\}$ where $cost(B_v|C_{t^*})$ is minimized. By Theorem A.5, $cost(B_v|C_v) \leq (1+220\varepsilon)cost(B_v|C_{t^*})$.

If $C_v^*$ is the set of good nodes that are not lost in $C_{ref}(v)$, then by Lemma 3.6 all the nodes in $C_v^*$ are correctly assigned to $v$, and so they are in $B_v$. Note that $A$ clusters the node in $B_v$ as follows: put all the nodes in $C_v^*$ in one cluster and make all the nodes in $B_v \setminus C_v$ singleton. By Lemma 3.7, we have that $cost(B_v|C_{t^*}) \leq \frac{4}{1-2\varepsilon} cost(B_v|C_v^*)$, so $cost(B_v|C_v) \leq \frac{4(1+220\varepsilon)}{(1-2\varepsilon)} cost(B_v|C_v^*) \leq 4(1+230\varepsilon)cost(B_v|C_v^*)$ since $\varepsilon \leq 1/1000$.

**Case 2: $C_{ref}(v)$ is a bad or poor cluster.** We know that all the nodes in $C_{ref}(v)$ are singletons in $A$. Let $t^*$ be the threshold in $\{1, (1+\epsilon), \ldots, (1+\epsilon)^{\lceil \log n \rceil + 1}\}$ where $cost(B_v|C_{t^*})$ is minimized. So $cost(B_{t^*}|C_v)$ is at most the cost of making $B_v$ singleton, i.e. $cost(B_v|C_{(1+\epsilon)^{\lceil \log n \rceil + 1}}) = cost(B_v|\emptyset)$. By Theorem A.5 $cost(B_v|C_v) \leq (1 + 220\varepsilon)cost(B_v|C_{t^*})$. So $cost(B_v|C_v) \leq (1+220\varepsilon)cost(B_v|\emptyset)$.

Finally, note that a node not in any $B_v$ is a singleton in both $B$ and $A$. Putting the above two cases together, So we have

$\mathbb{E}(cost(B)) \leq 4(1 + 50\epsilon)\mathbb{E}(cost(A)) \leq 4(1 + 230\epsilon)(1 + 8\epsilon)\mathbb{E}(cost(ref)) \leq 4(1 + 1000\epsilon)\mathbb{E}(cost(ref))$, where the last inequality uses the fact that $\epsilon < 1/1000$.

## 4. Random deletions

Let $n_0$ be the number of nodes in the graph just after the last recomputation. Our algorithm for deletions is quite simple:

1. Ignore deletions.

2. After $\varepsilon n_0/6$ updates, counting both insertions and deletions, recompute the clustering from scratch by treating all non-deleted nodes as if they had been inserted one by one again. These insertions are processed by Algorithm 3. Our recomputing procedure is described in Appendix B.

Recall that on a deletion update, a node to be deleted is chosen uniformly at random among the existing nodes. Observe that the choice of deletions is independent of the randomness used by our algorithm. At time $t$ of the algorithm, let $D_t$ be the nodes deleted since the last clustering recomputation. We think of $D_t$ as the nodes waiting to be deleted.

By construction, we have $|D_t| \leq \varepsilon n_0/6$. Since after the recomputation there are $n_0$ nodes in the graph, $D_t$ is a subset of (at least) $n_0$ nodes, and hence $\Pr(u \in D_t) \leq |D_t|/n_0 \leq \varepsilon/6$. There is inequality instead of equality, as the $\varepsilon n_0/6$ updates might contain insertions, resulting in a reduced probability of a node appearing in $D_t$.

Let $\mathcal{C}_{\text{NO-DEL}}$ be the clustering obtained by 5-approximate REFERENCE CLUSTERING at time $t$ where deletions $D_t$ are ignored. Let $\mathcal{C}_{\text{DEL}}$ be the clustering obtained by REFERENCE CLUSTERING at time $t$ in which deletions $D_t$ are considered. Let $P_t \subseteq V \times V$ be the node pairs that $\mathcal{C}_{\text{NO-DEL}}$ pays for. We aim to lower-bound the expected number of pairs in $P_t$ that $\mathcal{C}_{\text{DEL}}$ also pays for. Consider a pair $e = \{u, v\} \in P_t$. We will lower-bound the probability that the clustering of $u$ and $v$ is the same in $\mathcal{C}_{\text{DEL}}$ as in $\mathcal{C}_{\text{NO-DEL}}$

Consider a node $u$; the exact same analysis applies to $v$.

First, assume that $u$ is a singleton in $\mathcal{C}_{\text{NO-DEL}}$. This implies that the neighbor of $u$ with smallest rank, $w$, is not a pivot. Node $w$ is not a pivot because it has a neighbor $w'$ with a rank smaller than $w$. If none of $w, w'$, or $u$ is in $D_t$, then $u$ is a singleton in $\mathcal{C}_{\text{DEL}}$. Since we have $\Pr(w \in D_t \text{ or } w' \in D_t \text{ or } u \in D_t) \leq \Pr(w \in D_t) + \Pr(w' \in D_t) + \Pr(u \in D_t) \leq \varepsilon/2$, then in this case, the clustering of $u$ in $\mathcal{C}_{\text{DEL}}$ is the same as in $\mathcal{C}_{\text{NO-DEL}}$ with probability at least $1 - \varepsilon/2$.

Second, assume that $u$ is not a singleton in $\mathcal{C}_{\text{NO-DEL}}$. This implies that the highest-rank neighbor $w$ of $u$ is a pivot. By deleting nodes, and unless $w$ is deleted, $w$ remains a pivot. So, unless $u$ or $w$ are in $D_t$, the clustering of $u$ is the same in $\mathcal{C}_{\text{DEL}}$ and $\mathcal{C}_{\text{NO-DEL}}$. Since we have $\Pr(w \in D_t \text{ or } u \in D_t) \leq$

$\varepsilon/3$, in this case, the clustering of $u$ in $\mathcal{C}_{\text{DEL}}$ is the same as in $\mathcal{C}_{\text{NO-DEL}}$ with probability at least $1 - \varepsilon/3$.

This analysis implies that the clustering of a pair $\{u, v\}$ is the same in $\mathcal{C}_{\text{DEL}}$ as in $\mathcal{C}_{\text{NO-DEL}}$ with probability at least $1 - 5\varepsilon/6$. Hence, by the linearity of expectation,

$$\mathbf{E}\left[\text{cost}(\mathcal{C}_{\text{DEL}})\right] \geq (1 - 5\varepsilon/6)|P_t| \geq (1 - \varepsilon)\,\text{cost}(\mathcal{C}_{\text{NO-DEL}}).$$

## 5. Experiments

In this section, we empirically demonstrate that our approximation guarantee is better than that of REFERENCE CLUSTERING and the algorithm of (Cohen-Addad et al.). In the rest, we use DYNAMIC AGREEMENT to refer to the approach in (Cohen-Addad et al.).

**Algorithm Parameters** In SPARSE-PIVOT, for the BREAK-CLUSTER and UPDATE-CLUSTER subroutines we do the following: in BREAK-CLUSTER we consider $O(\log n)$ many candidates for $C_v$, estimate their costs and pick one with the lowest cost. In UPDATE-CLUSTER we update our $O(\log n)$ estimates by adding the new node, and again pick the one with lowest cost. To simplify the code, we heuristically alter the BREAK-CLUSTER and UPDATE-CLUSTER subroutines as follows. In BREAK-CLUSTER, for each node $u \in B_v$, we sample $O(\log n)$ nodes in $B_v$. If $u$ is attached to half of them, we add $u$ in $C_v$. In UPDATE-CLUSTER, we add $u$ to $C_v$, even though $u$ might not be attached to many nodes in $C_v$. After at least $\varepsilon|B_v|$ nodes are added to $B_v$, we rerun BREAK-CLUSTER. Note that the reason we get a $(20 + O(\varepsilon))$ approximation instead of a $(5 + O(\varepsilon))$ approximation is the BREAK-CLUSTER subroutine, so depending on the application, one can replace this subroutine with a version that one sees fit. Furthermore, we run RECOMPUTE every time the number of deletions reaches $\varepsilon N$, instead of the total number of updates. We observe that this does not degrade the approximation guarantee and slightly improves the running time.

We set the experiment parameters and the parameters for DYNAMIC-AGREEMENT to be the same as in (Cohen-Addad et al.). We choose a random ordering for the arrival of the nodes, and at each step, with probability $0.8$, we insert the next node, and with probability $0.2$, we delete a random node. If all the nodes have been inserted once, we delete them until no node is left. We set the parameter $\varepsilon$ for SPARSE-PIVOT to be $0.1$.

**Datasets** We use the same datasets as in (Cohen-Addad et al.) for a complete comparison. We evaluate the algorithms on two types of graphs.
(1) Sparse real-world graphs from SNAP (Jure, 2014): a social network (musae-facebook), an email network (email-Enron), a collaboration network (ca-AstroPh), and a paper citation network (cit-HepTh)

(2) The drift dataset (Vergara et al., 2012; Rodriguez-Lujan et al., 2014) from ICO Machine Learning Repository (Dua et al., 2017), which includes 13,910 points embedded in a space of 129 dimensions. A graph is constructed by placing an edge between two nodes if their Euclidean distance is less than a certain threshold. This setup is used to easily change the density of the graph and test how it affects the algorithms. The thresholds we choose are the same as in (Cohen-Addad et al.), and they are the mean of the distances between all nodes divided by $c \in \{10, 15, 20, 25, 30\}$. The lower the threshold, the sparser the graph. The density of a graph is the ratio of the number of edges and the number of nodes.

**Baselines** We use three baselines: making all nodes singletons, which we call SINGLETONS, DYNAMIC-AGREEMENT, and REFERENCE CLUSTERING. We divide the cost of each algorithm by the cost of SINGLETONS. Since REFERENCE CLUSTERING handles only node insertions, we process deletions in a way similar to (Cohen-Addad et al.). Note that our results *slightly* differ from that of (Cohen-Addad et al.) since they depend on the randomness of node arrivals. Moreover, the running time depends on the machine in which the algorithm is being run. Nevertheless, the scale of results we obtain does reproduce that of (Cohen-Addad et al.).

**Results: Approximation Guarantee** For all the datasets, our approximation guarantee is better than DYNAMIC-AGREEMENT and SINGLETONS. For SNAP graphs, we plot the correlation clustering objective every 50 steps. Figure 1 shows this objective for one of these graphs, and the rest can be found in Appendix F. The average clustering

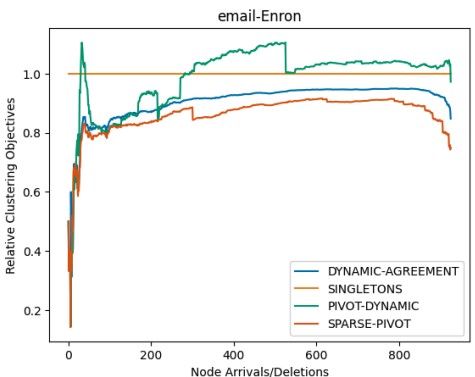

*Figure 1.* Comparison of the correlation clustering objective across the four algorithms. The lower the plot, the better.

objective for the drift dataset graphs is shown in Table 1.

**Results: Running time** Our experiments focus on the solution quality of SPARSE-PIVOT. Nevertheless, we compare the running times for completeness and illustrate that SPARSE-PIVOT is faster than DYNAMIC-AGREEMENT in practice, see Appendix F.

| Density | DA | RC | SP |
|---------|------|------|------|
| 235.36 | 0.69 | 0.59 | 0.6 |
| 114.87 | 0.59 | 0.64 | 0.49 |
| 69.74 | 0.5 | 0.5 | 0.41 |
| 52.17 | 0.39 | 0.42 | 0.32 |
| 42.25 | 0.35 | 0.35 | 0.29 |

*Table 1.* Clustering Objective of DYNAMIC-AGREEMENT (DA), REFERENCE CLUSTERING (RF) and SPARSE-PIVOT (SP). The smaller the number, the better.

## Acknowledgements

K. Makarychev was supported by the NSF Awards CCF-1955351 and EECS-2216970.

S. Mitrović was supported by the NSF Early Career Program No. 2340048 and the Google Research Scholar Program.

## Impact Statement

This paper presents work that aims to advance algorithmic tools for data partitioning, a method used in the field of Machine Learning. There are many potential societal consequences of our work, none of which we feel must be specifically highlighted here.

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

# A. Implementing cost estimates

Our clustering procedures, e.g., Algorithm 3, for each pivot $v$ maintain two sets of nodes: $B_v$ and $C_v$. The set $B_v$ is a set of nodes whose pivot is $v$. However, having $B_v$ as one cluster might sometimes be very far from an optimal clustering of the nodes within $B_v$. So, our algorithm computes a cluster $C_v \subseteq B_v$ for which we can guarantee a relatively low cost; details of this analysis are provided in our proof of Theorem 2.1. To compute a cluster $C_v$, our algorithm estimates the costs of several clusters and chooses $C_v$ as the cluster with the lowest estimated cost. In this section, we describe how to estimate the cost of a cluster efficiently, that is, in only $\mathrm{poly}(\log n, 1/\varepsilon)$ time. We need to handle two cases: how to estimate the cost of a given cluster $C$ from scratch, i.e., in a static manner, and how to maintain the cost estimate of a cluster $C$ under node insertions.

We need the former case for our recomputation or when we create an entirely new $B_v$ because $v$ is just becoming a pivot. It might be tempting to create new $B_v$ by "pretending" that the nodes of $B_v$ have been inserted one by one. However, this approach has a small subtlety. Namely, when a node $v$ is inserted, only at that point are the edges incident to $v$ included in our graph, and no edge between $v$ and a node inserted in the future is known. On the other hand, if we "pretend" that an already existing sequence of nodes is just now inserted, then a currently processed node also has edges to its neighbors that have yet to be processed/inserted. This scenario slightly affects how we count edges and non-edges within $B_v$ or $C_v$.

## A.1. Static version

### A.1.1. WITHIN-CLUSTER COST ESTIMATE

We first design a procedure to estimate the cost within a cluster $C$, i.e., the number of non-edges within $C$, by spending only $O(\log n)$ time per edge. It is provided as Algorithm 4. As shown by Lemma A.1, this estimate is tightly concentrated as

---

**Algorithm 4** IN-CLUSTER-COST-ESTIMATE

1: **Input** set $C \subseteq V$
2: $\tau_C \overset{\text{def}}{=} 5 \cdot |C| \cdot \log(n)/\varepsilon^3$
3: **for** $i = 1 \ldots \tau_C$:
4:     Uniformly at random, sample two distinct nodes $v$ and $w$ from $C$
5:     **if** $\{w, v\}$ is a non-edge, then $S \leftarrow S + 1$
6: **return** $S \cdot \binom{|C|}{2}/\tau_C$

---

long as the number of non-edges is in $\Omega(|C|)$. If the number of non-edges is lower, then their actual number is irrelevant to our algorithm. Updating this cost dynamically is more involved, and we elaborate on details in Appendix A.2.

**Lemma A.1** (In-cluster cost estimate). *Let $C \subseteq V$ be a set of nodes. Then, for $\varepsilon < 1/2$, Algorithm 4 (IN-CLUSTER-COST-ESTIMATE) uses $O(|C| \log(n)/\varepsilon^2)$ running time and outputs $Y$ for which with high probability the following holds:*

- *If the number of non-edges within $C$ if at least $2\varepsilon|C|$, then $Y$ is a $1 \pm \varepsilon$ multiplicative approximation of that number of non-edges.*

- *Otherwise, $Y < 3\varepsilon|C|$.*

*Proof.* Let $t$ be the number of non-edges in $C$. Let $X_i$ be a random $0/1$ variable equal $1$ iff the $i$-th $\{v, w\}$ pair sampled by Algorithm 4 is a non-edge. Then,

$$\mathbf{E}\left[X_i\right] = \Pr\left(X_i = 1\right) = \frac{t}{\binom{|C|}{2}}.$$

Let $S'$ be the value of $S$ at the end of Algorithm 4. Since $S' = \sum_{i=1}^{\tau_C} X_i$, we have

$$\mathbf{E}\left[S'\right] = \tau_C \cdot \frac{t}{\binom{|C|}{2}}. \tag{1}$$

Let

$$Y \overset{\text{def}}{=} \frac{S' \cdot \binom{|C|}{2}}{\tau_C}$$

be the output of Algorithm 4. Observe that $\mathbf{E}[Y] = t$. Therefore, the expected value of $Y$ is the desired one. In the rest, we analyze the concentration bounds of this estimator.

Consider two cases based on the value of $t$.

**Case $t \geq 2\varepsilon|C|$.** Recall that $\tau_C = 5|C|\log(n)/\varepsilon^3$. Replacing the bounds on $t$ and $\tau_C$ in Equation (1) yields

$$\mathbf{E}[S'] \geq \frac{10|C|^2 \cdot \log n}{\varepsilon^2 \binom{|C|}{2}} \geq \frac{20\log n}{\varepsilon^2}. \tag{2}$$

Since $S'$ is a sum of independent $0/1$ random variables, by the Chernoff bound, it holds that[2]

$$\Pr\left(|S' - \mathbf{E}[S']| < \varepsilon\mathbf{E}[S']\right) \leq n^{-6}.$$

This now implies that for $t \geq 2\varepsilon|C|$, with probability at least $1 - n^{-6}$, $Y$ is a $(1 \pm \varepsilon)$ multiplicative approximation of $t$.

**Case $t < 2\varepsilon|C|$.** In this case, we would like to claim that very likely it holds that $Y < 3\varepsilon|C|$. This can be argued by applying the Chernoff bound as follows.

Observe that for this value of $t$ we have

$$\mathbf{E}[S'] < \frac{20\log n}{\varepsilon^2}.$$

Hence,

$$\Pr\left(S' > (1 + \varepsilon)\frac{20\log n}{\varepsilon^2}\right) \leq n^{-6}.$$

Therefore, with probability at least $1 - n^{-6}$, it holds that

$$Y \leq \frac{(1 + \varepsilon)\frac{20\log n}{\varepsilon^2} \cdot \binom{|C|}{2}}{\tau_C} < (1 + \varepsilon)2|C| < 3\varepsilon|C|,$$

for $\varepsilon < 1/2$. □

### A.1.2. SINGLE-CLUSTER + SINGLETONS COST ESTIMATE

We now discuss how to estimate the cost of $B$ for a given $C$, where $C$ is taken as a single cluster, while all the nodes in $B - C$ are singletons. By cost of $B$ we mean the number of edges with at least one endpint in $B - C$ and the other endpoint in $B$, plus the number of non-edges in $C$. Note that the true correlation clustering cost of $B$ is the above cost plus half of the edges from $B$ to outside of $B$, but since we need these costs to compare different choices of $B$ and the number of edges going outside of $B$ is indipendent of this choice, it does not influence our comparison.

In this section, given two node subsets $X$ and $Y$, we use $e(X, Y)$ to denote the number of edges with one endpoint in $X$ and the other in $Y$. In particular, $e(X, X)$ is the number of edges in $G[X]$. We abbreviate $e(X, X)$ to $e(X)$.

First, the entire cost of $C$, denoted by $cost(C)$, equals the sum of $e(C, V - C)$ and the number of non-edges within $C$. Observe that

$$e(C) = \binom{|C|}{2} - [\text{the number of non-edges within } C].$$

So, we have

$$\sum_{w \in C} d(w) = e(C, V - C) + 2e(C) = e(C, V - C) + 2\binom{|C|}{2} - 2 \cdot [\text{the number of non-edges within } C].$$

This now implies that

$$cost(C) = \sum_{w \in C} d(w) - 2\binom{|C|}{2} + 3 \cdot [\text{the number of non-edges within } C].$$

---

[2]The constant $-6$ can be made arbitrarily large by increasing the constant in $\tau_C$.

Second, it remains to account for making the nodes in $B - C$ singletons. The edges $E(C, B - C)$ are already accounted for by $\sum_{w \in C} d(w)$. To account for the cost of making $B - C$ singletons, the following procedure can be used:

- For each node $w \in B - C$, iterate over all the edges in its adjacency list and

- to an edge from $E(w, C)$ assign weight 0; to an edge from $E(w, B - C)$ assign weight $1/2$; and to an edge from $E(w, V - B)$ assign weight 1.

Observe that an edge $\{x, y\}$ with $x, y \in B - C$ is counted twice: once in the adjacency list of $x$ and once in the adjacency list of $y$. Hence, the sum of these edge weights and $cost(C)$ equals the cost of clustering $B$.

However, using the above procedure directly can result in a running time that is too long. Instead, we would like a procedure with the running time of $O(|B| \cdot \mathrm{poly}(\log n, 1/\varepsilon))$. Nevertheless, estimating the sum of the edge weights in the desired time is simple. We outline one such approach in Algorithm 5.

---

**Algorithm 5** COST-ESTIMATE

1: **Input** node sets $B$ and $C \subseteq B$.
2: $\widetilde{cost} = \sum_{w \in C} d(w) - 2\binom{|C|}{2} + 3 \cdot$ IN-CLUSTER-COST-ESTIMATE$(C)$
3: Let $\eta \overset{\text{def}}{=} 10 \cdot \log(n)/\varepsilon^3$
4: **for** $w \in B - C$:
5:     Sample $\eta$ edges incident to $w$, each edge sampled independently and uniformly at random
6:     For $S \in \{C, B - C, V - B\}$, let $Z_S(w)$ be $d(w)/\eta$ multiplied by number of sampled edges incident to $S$
7:     $\widetilde{cost} = \widetilde{cost} + \frac{Z_{B-C}(w)}{2} + Z_{V-B}(w)$
8: **return** $(\widetilde{cost} + 9\varepsilon|C|)/(1 - 37\varepsilon)$

---

Let $Z_S(w)$ be as defined in Algorithm 5. Observe that $\mathbf{E}[Z_S(w)] = e(w, S)$. A straightforward analysis, and identical to that presented in the proof of Lemma A.1, shows that for $e(w, S) \geq 2\varepsilon d(w)$ the value of $Z_S(w)$ computed in Algorithm 5 is with high probability a $(1 \pm \varepsilon)$ factor approximation of $e(w, S)$. For $e(w, S) < 2\varepsilon d(w)$, the same analysis yields that with high probability, it holds that $Z_S(w) < 3\varepsilon d(w)$. Hence, when $e(w, S) < 2\varepsilon d(w)$, the estimate $Z_S(w)$ is not necessarily within $1 \pm \varepsilon$ factor of its expected value, and thus the error has to be accounted for differently. Next, we explain how to account for it.

Trivially, at least one among $e(w, C)$, $e(w, B - C)$, and $e(w, V - B)$ is at least $d(w)/3 > 2\varepsilon d(w)$, for $\varepsilon < 1/6$. If $e(w, C) \geq d(w)/3$, we charge each $Z_S(w) < 3\varepsilon d(w)$ to the cost of $E(w, C)$ paid by $\sum_{w \in C} d(w)$. This incurrs an extra cost of at most $(2 \cdot 3\varepsilon d(w))/(d(w)/3) = 18\varepsilon$ per an edge in $E(w, C)$. The analogous analysis applies to the case $e(w, V - B) \geq d(w)/3$ and $e(w, B - C) \geq d(w)/3$. The only difference is that $Z_{B-C}(w)$ is divided by 2 in Algorithm 5, so for that case, the analysis yields a $36\varepsilon$ increased cost per edge.

Overall, this analysis implies that, with high probability, $\widetilde{cost}$ in COST-ESTIMATE is a $(1 \pm 37\varepsilon)$ multiplicative and $9\varepsilon|C|$ additive approximation of the cost of clustering $B$. The additive approximation comes from Lemma A.1 and the fact that $3 \cdot$ IN-CLUSTER-COST-ESTIMATE$(C)$ figures in the output of COST-ESTIMATE.

**Lemma A.2** (Cost estimate of single-cluster + singletons)**.** *Let $\varepsilon < 1/111$. Given two node sets $B$ and $C \subseteq B$, let $cost^*(B|C)$ be the cost of clustering $B$ in which $C$ is a single cluster and $B - C$ are singletons, which is defined to be the number non-edges in $C$ plus the number of edges in $B$ with at least one endpoint in $B - C$. Then, if COST-ESTIMATE$(B, C)$ (Algorithm 5) outputs $X$, we have $cost^*(B|C) \leq X \leq (1 + 111\varepsilon)cost^*(B|C) + 27\varepsilon|C|$. Moreover, the algorithms run in $O(|B| \cdot \log(n)/\varepsilon^3)$ time.*

Note that since $(1 - 37\varepsilon)cost^*(B|C) - 9\varepsilon|C| \leq \widetilde{cost} \leq (1 + 37\varepsilon)cost^*(B|C) + 9\varepsilon|C|$ and $X = (\widetilde{cost} + 9\varepsilon|C|)/(1 - 37\varepsilon)$, and $\varepsilon < 1/111$ we have that $cost^*(B|C) \leq X \leq (1 + 111\varepsilon)cost^*(B|C) + 27\varepsilon|C|$.

### A.1.3. COST COMPARISON

Let the estimate that Algorithm 5 makes for $cost^*(B|C)$ be $\widetilde{cost}(B|C)$. We show that $\widetilde{cost}(B|C)$ is a good enough measure for choosing a $C$ with low $cost(^*B|C)$.

**Lemma A.3.** *If $cost^*(B|C) \geq |C|/4$ then $cost^*(B|C) \leq \widetilde{cost}(B|C) \leq (1 + 219\varepsilon)cost^*(B|C)$.*

---

**Algorithm 6** COST-COMPARISON

1: Use Algorithm 5 to compute $\widetilde{cost}(B|C)$ and $\widetilde{cost}(B|C')$.
2:   **if** $\widetilde{cost}(B|C) < \widetilde{cost}(B|C')$:
3:       **return** $C$
4:   **else**:
5:       **return** $C'$.

---

*Proof.* Since $|C| \geq 4cost(B|C)$, by Lemma A.2, we have that $\widetilde{cost}(B|C) \leq (1 + 111\varepsilon)cost^*(B|C) + 27\varepsilon|C| \leq (1 + 111\varepsilon)cost^*(B|C) + 108\varepsilon cost^*(B|C) \leq (1 + 219\varepsilon)cost^*(B|C)$. $\square$

**Lemma A.4.** *If $cost^*(B|C) \leq |C|/4$, then for any $C'$ such that $C' \subset C$ or $C \subset C'$, we have $cost^*(B|C) < cost^*(B|C')$ and $\widetilde{cost}(B|C) \leq \widetilde{cost}(B|C')$.*

*Proof.* First suppose that $C' \subset C$. Take a node $u \in C \setminus C'$, we know that $cost(B|C) \geq |C| - 1 - d_C(u) \geq |C|/2 - d_C(u)$. So $|C|/2 \leq d_C(u)$. Now we have $cost^*(B|C') \geq d_C(u) \geq |C|/2 > cost^*(B|C)$. Similarly, suppose $C \subset C'$. Take a node $u \in C' \setminus C$. We know that $cost^*(B|C) \geq d_C(u)$, so $d_C(u) \leq |C|/4$. Moreover, $cost^*(B|C') \geq |C| - 1 - d_C(u) > |C|/2$. So in both cases $cost^*(B|C') > |C|/2 > cost^*(B|C)$.

Furthermore, by Lemma A.2 $\widetilde{cost}(B|C) \leq (1 + 111\varepsilon)|C|/4 + 27\varepsilon|C|$ and by Lemma A.3 we have $\widetilde{cost}(B|C') \geq (1 + 219\varepsilon)cost^*(B|C') \geq (1 + 219\varepsilon)|C|/2$. So we have $\widetilde{cost}(B|C') < \widetilde{cost}(B|C)$. $\square$

We use Algorithm 6 to develop Algorithm 7 that finds a dense cluster $C_v$ inside $B_v$, where $B_v$ is the set of all the nodes that are assigned to pivot $v$. Recall that $C_t$ is the set of nodes in $B_v$ with degree at least $t$.

---

**Algorithm 7** BREAK-CLUSTER

1: **Input** Set $B_v$.
2: For any $t > 0$, let $C_t = \{u \in B_v, d(u) \geq t\}$.
3: initialize $t_v = 0, C_v = B_v$.
4: **for** $t \in \{1, (1 + \epsilon), (1 + \epsilon)^2, \ldots, (1 + \epsilon)^{\lceil \log n \rceil}\}$:
5:    $C_v \leftarrow$ COST-COMPARISON$(B_v, C_t, C_v)$.
6: **return** $C_v$.

---

For any $C \subseteq B$, let $cost(B|C)$ be the cost of making $C$ a cluster, and $B - C$ singletons. This cost is equal to half the number of edges with exactly one endpoint in $B$, plus the number of edges with one endpoint in $B - C$ and the other endpoint in $B$, plus the number of non-edges in $C$. In fact, $cost(B|C)$ is $cost^*(B|C)$ plus half the number of edges that leave $B$.

**Theorem A.5.** *Let $t^*$ be the threshold among $1, (1 + \varepsilon), \ldots, (1 + \varepsilon)^{\lceil \log n \rceil}$ where $cost(B_v|C_{t^*})$ is minimized. If Algorithm 7 returns $C_{\tilde{t}}$, then $cost(B_v|C_{\tilde{t}}) \leq (1 + 219\varepsilon)cost(B_v|C_{t^*})$.*

*Proof.* First note that since $cost(B_v|C_t)$ is $cost^*(B_v|C_t)$ plus half the number of edges that leave $B_v$, for any $t, t'$ we have $cost(B_v|C_t) - cost(B_v|C_{t'}) = cost^*(B_v|C_t) - cost^*(B_v|C_{t'})$, and thus we can use $cost^*$ for comparing the costs, i.e. $t^*$ minimizes $cost^*(B_v|C_t)$ as well as $cost(B_v|C_t)$.

We prove the Theorem by induction: Suppose that for any $j$, $t_j$ is the threshold among $1, (1 + \varepsilon), \ldots, (1 + \varepsilon)^j$ such that $cost^*(B_v|C_{t_j})$ is minimized, and the output of the for loop in Algorithm 7 for $t \in \{1, (1 + \varepsilon), \ldots, (1 + \varepsilon)^j\}$ is $C_{\tilde{t}_j}$. Fix some $i$. Suppose that $cost^*(B_v|C_{\tilde{t}_i}) \leq (1 + 219\varepsilon)cost^*(B_v|C_{t_i})$. Note that $C_{\tilde{t}_{i+1}} = $ COST-COMPARISON$(B_v, C_{(1+\varepsilon)^{i+1}}, C_{\tilde{t}_i})$. We show that $cost^*(B_v|C_{\tilde{t}_{i+1}}) \leq (1 + 219\varepsilon)cost^*(B_v|C_{t_{i+1}})$.

For ease of notation let $C_1 = C_{t_i}$, $C_2 = C_{t_{i+1}}$, and $C' = C_{(1+\varepsilon)^{i+1}}$. So we have $C_2 = \arg\min_{C \in \{C_1, C'\}} cost^*(B_v|C)$. Let $\tilde{C}_1 = C_{\tilde{t}_i}$ and $\tilde{C}_2 = C_{\tilde{t}_{i+1}}$. Assuming that $cost^*(B_v|\tilde{C}_1) \leq (1 + 219\varepsilon)cost^*(B_v|C_1)$, we need to show that $cost^*(B_v|\tilde{C}_2) \leq (1 + 219\varepsilon)cost^*(B_v|C_2)$, where $\tilde{C}_2 = $ COST-COMPARISON$(B_v, C', \tilde{C}_1)$.

Note that $C' \subseteq \tilde{C}_1$. This is because $\tilde{C}_1 = C_{\tilde{t}_i}$ and $C' = C_{(1+\varepsilon)^{i+1}}$ where $\tilde{t}_i \leq (1 + \varepsilon)^i < (1 + \varepsilon)^{i+1}$. We prove the rest of the theorem in the following cases.

**Case 1:** $cost^*(B|C') < |C'|/4$. In this case, by Lemma A.4 we know that $cost^*(B|C') < cost^*(B|\tilde{C}_1)$ and $\widetilde{cost}(B|C') < \widetilde{cost}(B|\tilde{C}_1)$. So $\tilde{C}_2 = C'$. By induction hypothesis, we have $cost^*(B|C') < cost^*(B|\tilde{C}_1) < (1 + 219\varepsilon)cost^*(B|C_1)$. Since $cost^*(B|C') < (1 + 219\varepsilon)cost^*(B|C')$ and $C_2 \in \{C', C_1\}$, we have $cost^*(B|\tilde{C}_2) = cost^*(B|C') \leq (1 + 219\varepsilon)cost^*(B|C_2)$.

**Case 2:** $cost^*(B|\tilde{C}_1) < |\tilde{C}_1|/4$: Similar to case 1, in this case by Lemma A.4 we know that $cost^*(B|\tilde{C}_1) < cost^*(B|C')$ and $\widetilde{cost}(B|\tilde{C}_1) < \widetilde{cost}(B|C')$. So $\tilde{C}_2 = \tilde{C}_1$. By induction hypothesis, we have $cost^*(B|\tilde{C}_1) < (1 + 219\varepsilon)cost^*(B|C_1)$. Since $cost^*(B|\tilde{C}_1) < (1 + 219\varepsilon)cost^*(B|C')$ and $C_2 \in \{C', C_1\}$, we have $cost^*(B|\tilde{C}_2) = cost^*(B|\tilde{C}_1) \leq (1 + 219\varepsilon)cost^*(B|C_2)$.

**Case 3:** $cost^*(B|C') \geq |C'|/4$ **and** $cost^*(B|\tilde{C}_1) > |\tilde{C}_1|/4$: In this case, by Lemma A.3 we have that $\widetilde{cost}(B|C')$ and $\widetilde{cost}(B|\tilde{C}_1)$ are $(1 + 219\varepsilon)$ approximations of $cost^*(B|C')$ and $cost^*(B|\tilde{C}_1)$.

If $\widetilde{cost}(B|C') < \widetilde{cost}(B|\tilde{C}_1)$, then we have $\tilde{C}_2 = C'$. If $C_2 = C'$, then we are done. So assume that $C_2 = C_1$, which means that $cost^*(B|C_1) < cost^*(B|C')$. So we have $cost^*(B|C') \leq \widetilde{cost}(B|C') \leq \widetilde{cost}(B|\tilde{C}_1) \leq (1 + 219\varepsilon)cost^*(B|C_1)$ where the last inequality comes from the induction hypothesis. So we have $cost^*(B|\tilde{C}_2) \leq (1 + 219\varepsilon)cost^*(B|C_2)$

Similarly if $\widetilde{cost}(B|C') > \widetilde{cost}(B|\tilde{C}_1)$, then we have $\tilde{C}_2 = \tilde{C}_1$. If $C_2 = C_1$, then we are done by induction hypothsis. So assume that $C_2 = C'$, which means that $cost^*(B|C_1) > cost^*(B|C')$. So we have $cost^*(B|\tilde{C}_1) \leq \widetilde{cost}(B|\tilde{C}_1) < \widetilde{cost}(B|C') \leq (1 + 219\varepsilon)cost^*(B|C')$. So $cost^*(B|\tilde{C}_2) \leq (1 + 219\varepsilon)cost^*(B|C_2)$.

$\square$

**Theorem A.6.** *Algorithm 7 runs in $|B_v|$ poly$(\log n, 1/\varepsilon)$ time.*

*Proof.* Note that Algorithm 7 calls Algorithm 6 $O(\log n)$ times. Each run of Algorithm 6 takes $|B_v|$ poly$(\log(n), 1/\varepsilon)$ time: this is because estimating $\widetilde{cost}(B|C_t)$ by Algorithm 5 for a set $C$ takes $O(|C| \log(n)/\varepsilon^3)$ time. $\square$

## A.2. Dynamic version: UPDATE-CLUSTER($B_v, u$)

The previous section describes how to estimate the cost of clustering $B$ in which $C \subseteq B$ is a cluster, while $B - C$ are singletons. When a node is inserted, we cannot afford to estimate these costs from scratch. Rather, we want to *update* the estimate based on the inserted node.

### A.2.1. INSERTING A NODE INTO $B - C$

Assume that we already have an estimate of the cost of clustering $B$ as guaranteed by Lemma A.2; let $\widetilde{cost}$ be that estimate. Assume that a node $z$ is inserted in $B - C$. Hence, $z$ will be a singleton. When $\widetilde{cost}$ was computed no edge incident to $z$ was in the graph. Thus, updating the cost estimate is easy in this case: the cost estimate of clustering $C$ together and $B - C + z$ as singletons equals $\widetilde{cost} + d(z)$.

### A.2.2. INSERTING A NODE INTO $C$

Assume that we already have an estimate of the cost of clustering $B$ as guaranteed by Lemma A.2; let $\widetilde{cost}$ be that estimate. Assume that a node $z$ is inserted in $C$. Let $\widetilde{cost}'$ be the new cost estimate we aim to obtain. Based on Line 2 of Algorithm 5, the difference $\widetilde{cost}' - \widetilde{cost}$ contains three components: $d(z)$; $-2\binom{|C|+1}{2} + 2\binom{|C|}{2}$; and, $3 \cdot$ IN-CLUSTER-COST-ESTIMATE$(C + z) - 3 \cdot$ IN-CLUSTER-COST-ESTIMATE$(C)$. The first two components can be updated in $O(1)$ time. Even $3 \cdot$ IN-CLUSTER-COST-ESTIMATE$(C)$ can be updated in $O(1)$ time – together with $\widetilde{cost}$, we store the value of IN-CLUSTER-COST-ESTIMATE$(C)$ that led to $\widetilde{cost}$ itself. So, it remains to describe how to obtain IN-CLUSTER-COST-ESTIMATE$(C + z)$ in poly$(\log n, 1/\varepsilon)$ time, as we do next.

Let $S_C$ be the value of $S$ at the end of IN-CLUSTER-COST-ESTIMATE$(C)$ invocation. Let $S_{C+z}$ be a value of $S$ corresponding to IN-CLUSTER-COST-ESTIMATE$(C + z)$ that we aim to obtain. We initialize $S_{C+z} = S_C$, and then update $S_{C+z}$ as follow.

First, given definition of $\tau_C$ on Line 2 of Algorithm 4, to compute $S_{C+z}$ we sample $5 \cdot \log(n)/\varepsilon^3$ pairs $(v, w) \in (C + z) \times (C + z)$, and for each non-edge $\{v, w\}$ we increment $S_{C+z}$ – the same as Algorithm 4 does.

However, the $(v, w)$ pairs sampled by Algorithm 4 to compute $S_C$ are simple from $C \times C$, while for $S_{C+z}$ we would like each pair to be sampled from $(C + z) \times (C + z)$. So, second, to account for that, we resample some of the pairs used in computing $S_C$. This also implies that while our algorithm estimates the cost of $C$ by Algorithm 4, it also stores in an array all the $\{v, w\}$ pairs it sampled within the for-loop. Let that array of samples be called $A_C$.

When computing $S_C$, a pair $\{v, w\}$ is sampled with probability $p_C = 1/\binom{|C|}{2}$. However, when computing $S_{C+z}$, that same pair is sampled with probability $p_{C+1} = 1/\binom{|C|+1}{2}$. Let $q = p_C/p_{C+1}$ So, in $A_C$, we resample a pair $\{v, w\}$ with probability $1 - q$, and otherwise, with probability $q$, $\{v, w\}$ is not resampled. With this process, we have that $\{v, w\} \in C \times C$ remains unchanged with probability $1/\binom{|C|+1}{2}$, as desired.

If a pair is resampled, then the new pair is $\{z, u\}$, where $u$ is a node from $C$ chosen uniformly at random. So, we have

$$\Pr(\{z, u\} \text{ is sampled}) = \frac{1}{|C|} \cdot \left(1 - \frac{\binom{|C|}{2}}{\binom{|C|+1}{2}}\right) = \frac{1}{|C|} \cdot \frac{2}{|C| + 1} = \frac{1}{\binom{|C|+1}{2}},$$

as we aim to achieve. But how many pairs are resampled? How does one choose which pairs to resample in $\mathrm{poly}(\log n, 1/\varepsilon)$ time?

The expected number of resampled pairs is

$$\mathbf{E}\left[|A_C| \cdot (1 - q)\right] = \tau_C \cdot \frac{2}{|C| + 1} = O(\log(n)/\varepsilon^3).$$

Hence, by a direct application of the Chernoff bound, with high probability, the number of resampled pairs from $A_C$ is $\mathrm{poly}(\log n, 1/\varepsilon)$.

The final piece elaborates on efficiently finding pairs to resample. Consider a process that iterates over the elements in $A_C$ and resamples each with probability $1 - q$. A downside is that this approach takes $\Theta(|C|)$ time, which is too slow for our goal. Instead, we observe that the index of *the first* element resampled in $A_C$ is drawn from the geometric distribution with parameter $1 - q$. This observation leads to the following efficient procedure for resampling elements from $A_C$:

1. Initialize $i = 0$.

2. Repeat while $i \leq |A_C|$:
    - sample an index $j$ from the Geometric distribution with parameter $1 - q$;
    - $i = i + j$;
    - if $i \leq |A_C|$, resample the $i$-th $\{v, w\}$ pair in $A_C$.

This latter approach enables us to spend time proportional to the number of resampled pairs – as opposed to $|A_C|$ – which we know is $\mathrm{poly}(\log n, 1/\varepsilon)$ with high probability.

**Lemma A.7** (Dynamic cost estimate of single-cluster + singletons). *There exists an algorithm that, on a node insertion, updates* COST-ESTIMATE$(B, C)$ *in* $\mathrm{poly}(\log n, 1/\varepsilon)$ *time with high probability. The approximation guarantees are the same as those stated in Lemma A.2.*

### A.2.3. UPDATE-CLUSTER$(B_v, u)$

Subroutine UPDATE-CLUSTER$(B_v, u)$ is essentially BREAK-CLUSTER with the costs of $C$ and $B - C$ updated dynamically as explained in Appendices A.2.1 and A.2.2.

More precisely, when BREAK-CLUSTER is invoked, we store all the estimates $\widetilde{cost}(B_v|C_t)$, the sets $B_v$ and $C_t$, for $t = 1, (1 + \epsilon), \ldots, (1 + \epsilon)^{\log n}$. Then, we use Lemma A.7 to update these costs when $u$ is inserted into $B_v$ in $\mathrm{poly}(\log n, \varepsilon)$ time. Note that if $d(u) \geq t$, then $u$ joins $C_t$, and otherwise $u$ joins $B_v \setminus C_t$. So each run of Algorithm 6 takes $\mathrm{poly}(\log n, 1/\varepsilon)$, and so UPDATE-COST$(B_v, u)$ works in $\mathrm{poly}(\log n, 1/\varepsilon)$ time.

## B. Recompute

Let $n_0$ be the number of nodes in the graph just after the last recompute. After $\varepsilon n_0/6$ updates, we perform a recomputation. Our RECOMPUTE procedure is as follows:

- Purge from the database soft-deleted nodes.

- Assign to each node $u$ a rank $\pi(u)$ chosen uniformly at random from $[0, 1]$.

- Initialize $p(u) = u$ for all nodes $u$.

- Sort the nodes in the increasing order with respect to $\pi$.

- Insert the nodes, one by one, in this sorted order. The insertions are handled by Algorithm 3, except that a new $\pi(u)$ value is not obtained within Algorithm 3, but is used the one computed in the first step of RECOMPUTE.

The nodes are processed in the ordering based on their $\pi$ values for the following reasons. Given a node $u$, Algorithm 3 performs updates or exploration only when ranks are smaller than $\pi(u)$. In particular, the node $v$ defined in that algorithm is used only if $\pi(v) \leq \pi(u)$.

The running time of RECOMPUTE is a constant factor of the running time used to process the insertions. To see that, charge a recomputation running time to the $\varepsilon n_0/6$ most recent updates. Observe that this kind of charge is applied to each update during only one RECOMPUTE. Hence, at most $n_0 + \varepsilon n_0/6 < 2n_0$ insertions are charged to $\varepsilon n_0/6$ updates. So, each update is charged $12/\varepsilon$ insertions. Since an insertion takes $\text{poly}(\log n, 1/\varepsilon)$ amortized time, this additional charge also takes $\text{poly}(\log n, 1/\varepsilon)$ time per update.

## C. Auxilary Lemmas

**Lemma C.1.** *Let $\alpha < 1$ be a constant. Let A be a clustering algorithm, where the probability of a node $v$ being a pivot over all orderings is at most $1/d(v)$. The expected sum of the degrees of $(A, \alpha)$-poor nodes is at most $4\alpha \frac{1+3/2 \cdot \alpha}{1-5/2 \cdot \alpha}$ times the total expected cost of A.*

*Proof.* Let $N_{low}(v)$ be the neighbors of $v$ with degree at most $\alpha d(v)$. Observe that $N_{low}(v)$ is independent of $\pi$ and $A$.

**The sum of poor-node degrees in a cluster.** First, we show that for any $\pi$ and $v$ such that $v$ is a pivot in $A$ wrt to $\pi$, the sum of the degrees of the $(A, \alpha)$-poor nodes in $v$'s cluster is at most $\alpha d(v) \cdot \min(3\alpha d(v), |N_{low}(v)|)$.

Fix a ordering $\pi$ of nodes where $v$ is a pivot in $A$. Note that if $N_{low}(v) = \emptyset$, then there are no $(A, \alpha)$-poor nodes in $v$'s cluster, and the sum of degrees of all its $(A, \alpha)$-poor nodes is zero. Hence, assume $N_{low}(v) \neq \emptyset$. The size of $v$'s cluster is at most $3\alpha d(v)$ since the cluster size is at most three times the degree of an $(A, \alpha)$-poor node. The latter is the case as an $(A, \alpha)$-poor node $u$ in a cluster $C$ is **not** light by definition. Hence, $d_C(u) > |C|/3$, which further implies $|C| < 3d_C(u) \leq 3d(u) \leq 3\alpha d(v)$, where $d(u) \leq \alpha d(v)$ be definition of $(A, \alpha)$-poor nodes. So, the sum of degrees of all $(A, \alpha)$-poor nodes in $v$'s cluster is at most $\alpha d(v) \cdot \min(3\alpha d(v), |N_{low}(v)|)$.

**The expected sum of poor-node degrees in $A$.** Let $Q^\pi$ be the set of $(A, \alpha)$-poor nodes wrt the ordering $\pi$. Let $Pivot_v$ be the event that $v$ is a pivot in $A$. We have

$$
\begin{aligned}
\mathbf{E}_\pi \left[ \sum_{u \in Q^\pi} d(u) \right] &= \sum_\pi \frac{1}{n!} \sum_{v \in P_{ref}^\pi} \sum_{u \in Q^\pi \cap C_v^\pi} d(u) \\
&= \sum_v \sum_{\pi \, : \, v \in P_{ref}^\pi} \frac{1}{n!} \sum_{u \in Q^\pi \cap C_v^\pi} d(u) \\
&\leq \sum_v \sum_{\pi \, : \, v \in P_{ref}^\pi} \frac{1}{n!} \cdot \alpha d(v) \cdot \min(3\alpha d(v), |N_{low}(v)|) \\
&= \sum_v \left( \alpha d(v) \cdot \min(3\alpha d(v), |N_{low}(v)|) \cdot \sum_{\pi \, : \, v \in P_{ref}^\pi} \frac{1}{n!} \right) \\
&= \sum_v \Pr(Pivot_v) \cdot \alpha d(v) \cdot \min(3\alpha d(v), |N_{low}(v)|).
\end{aligned}
$$

Note that $\Pr\left(Pivot_v\right) \leq \frac{1}{d(v)}$ in $A$. Hence, the expected sum of degrees of $(A, \alpha)$-poor nodes in $A$ is at most

$$\sum_v \alpha \cdot \min(|N_{low}(v)|, 3\alpha d(v)) = \sum_{v, N_{low}(v) \neq \emptyset} \alpha \cdot \min(|N_{low}(v)|, 3\alpha d(v)). \tag{3}$$

**Lower bounding the cost of $A$.** Now, we compare the above value to the cost of $A$. (The analysis we provide applies to the cost of any clustering, even the one incurred by the optimal solution.) Fix a ordering $\pi$. All the costs below are defined wrt $\pi$, and we avoid the superscript $\pi$.

Recall that for a vertex $w$, $C_A(w)$ denotes the cluster of $w$ in $A$. We define a cost function $\widehat{cost}_A$ to redistribute the cost $cost_A$ as follows

$$\widehat{cost}_A(w) \overset{\text{def}}{=} \frac{1}{2}\left(cost_A(w) + \frac{\sum_{u \in C_A(w)} cost_A(u)}{|C_A(w)|}\right).$$

In other words, in $\widehat{cost}_A$, a node $w$ distributes $1/2 \cdot cost_A(w)$ over the $|C_A(w)|$ nodes in $C_A(w)$, and $w$ itself pays for the remaining $1/2 \cdot cost_A(w)$. Note that $2 \cdot \widehat{cost}_A(w) \geq cost_A(w)$ and $\sum_w cost_A(w) = \sum_w \widehat{cost}_A(w)$.

Now we compute $\mathbf{E}\left[\widehat{cost}_A(v)\right]$ for $v$ with $N_{poor}(v) \neq \emptyset$. Note that we are not conditioning on $v$ being a pivot here. Let $t = |N_{poor}(v)|$, and let $t' = \min(t, 3\alpha d(v))$. We consider three cases based on $|C_A(v)|$:

- Case $|C_A(v)| \geq d(v) + \frac{t'}{2}$: then, there are at least $\frac{t'}{2}$ non-neighbors of $v$ in $C_A(v)$ and hence $cost_A(w) \geq \frac{t'}{2}$. Also, $2 \cdot \widehat{cost}_A(w) \geq cost_A(w) \geq \frac{t'}{2}$.

- Case $|C_A(v)| \leq d(v) - \frac{t'}{2}$: then, at least $\frac{t'}{2}$ neighbors of $v$ are outside of $C_A(v)$ and hence $cost_A(w) \geq \frac{t'}{2}$. Again, $2 \cdot \widehat{cost}_A(w) \geq cost_A(w) \geq \frac{t'}{2}$.

- Case $d(v) - \frac{t'}{2} \leq |C_A(v)| \leq d(v) + \frac{t'}{2}$: Let $t'' = |N_{low}(v) \cap C_A(v)|$. The number of non-neighbors of a node $u \in N_{poor}(v) \cap C_A(v)$ is least $|C_A(v)| - d(u) \geq d(v) - t'/2 - \alpha d(v) \geq (1 - 5/2 \cdot \alpha)d(v)$, where we used the fact that $t'/2 \leq 3/2 \cdot \alpha d(v)$. So

$$\sum_{u \in C_A(v)} cost_A(u) \geq \sum_{u \in N_{low}(v) \cap C_A(v)} cost_A(u) \geq t''(1 - 5/2 \cdot \alpha)d(v).$$

  Moreover, for each node $u \in N_{low}(v) \setminus C_A(v)$, $v$ incurs one unit of cost. Also, by definition, $t \geq |N_{low}(v)|$. Therefore, $cost_A(v) \geq t - t''$. Recall that this case assumes $|C_A(v)| \leq d(v) + t'/2 \leq d(v) + 3/2 \cdot \alpha d(v) = (1 + 3/2 \cdot \alpha)d(v)$. This yields

$$2 \cdot \widehat{cost}_A(v) \geq t - t'' + \frac{t''(1 - 5/2 \cdot \alpha)d(v)}{|C_A(v)|} \geq t - t'' + t''\frac{1 - 5/2 \cdot \alpha}{1 + 3/2 \cdot \alpha}. \tag{4}$$

  Observe that for $x \in [0, 1]$ and $t \geq t''$ we have $(1 - x)t \geq (1 - x)t''$, and hence $t - t'' + t''x \geq tx$. Thus, Equation (4) implies

$$2 \cdot \widehat{cost}_A(v) \geq t\frac{1 - 5/2 \cdot \alpha}{1 + 3/2 \cdot \alpha} \geq t'\frac{1 - 5/2 \cdot \alpha}{1 + 3/2 \cdot \alpha}.$$

Therefore, in any case, $\widehat{cost}_A(v) \geq \frac{1}{4}t'\frac{1-5/2\cdot\alpha}{1+3/2\cdot\alpha}$. So, the total cost of clustering $A$ is at least

$$\sum_{v, N_{poor}(v) \neq \emptyset} \frac{1}{4}\min(|N_{poor}(v)|, 3\alpha d(v))\frac{1 - 5/2 \cdot \alpha}{1 + 3/2 \cdot \alpha}.$$

Note that the above lower bound is deterministic; in particular, it holds for any randomness used by $A$. Comparing this lower bound to Equation (3), we conclude that the expected sum of degrees of poor nodes is at most $4\alpha\frac{1+3/2\cdot\alpha}{1-5/2\cdot\alpha}$ times the (expected) total cost of $A$, as advertised by the claim. $\qquad\square$

**Lemma C.2** (Cost of making light and heavy singletons). *Consider a fixed ordering $\pi$ and a clustering algorithm $A$. Let $A'$ be the algorithm that first runs $A$ and, after, makes a subset of $A$-light and $A$-heavy nodes singletons. The cost of $A'$ is at most $\frac{\beta+1}{\beta-1}$ the cost of $A$.*

*Proof.* For any algorithm $B$, let $d_{in}^B(u)$, $d_{out}^B(u)$ and $\bar{d}_{in}^B(u)$ be the number of neighbors of $u$ inside its cluster, number of neighbors of $u$ outside its cluster, and the number of non-neighbors of $u$ inside its cluster, respectively. Note that since $\pi$ is fixed, algorithm $B$ determines the clusters. Let $cost_B$ denote the cost of algorithm $B$ (with respect to $\pi$). We have $2 \cdot cost_B = \sum_u [d_{out}^B(u) + \bar{d}_{in}^B(u)]$. Let $L$ be the set of $A$-light nodes that $A'$ makes singletons, and $H$ be the set of $A$-heavy nodes that $A'$ makes singletons.

We can write the cost of algorithm $A'$ as follows:

$$2cost_{A'} = \sum_{u \in L \cup H} d(u) + \sum_{u \notin L \cup H} [d_{out}^{A'}(u) + \bar{d}_{in}^{A'}(u)] \leq \sum_{u \in L \cup H} [d(u) + d_{in}^A(u)] + \sum_{u \notin L \cup H} [d_{out}^A(u) + \bar{d}_{in}^A(u)]$$

where the inequality comes from the fact that going from $A'$ to $A$, any new cost on a node $u \notin L \cup H$ is due to an edge inside $u$'s cluster in $A$ that it attached to a node in $L \cup H$. We prove that $\sum_{u \in L \cup H}[d(u) + d_{in}^A(u)] \leq \frac{\beta+1}{\beta-1} \sum_{u \in L \cup H}[d_{out}^A(u) + \bar{d}_{in}^A(u)]$. Given this, we will have that $cost_{A'} \leq \frac{\beta+1}{\beta-1} cost_A$.

If $u$ is $A$-heavy, then $d(u) \geq \beta|C_A(u)| \geq \beta d_{in}^A(u)$. So $d_{out}^A(u) \geq (\beta - 1)d_{in}^A(u)$, and hence $d(u) + d_{in}^A(u) = d_{out}^A(u) + 2d_{in}^A(u) \leq \frac{\beta+1}{\beta-1}d_{out}^A(u)$.

If $u$ is a light node, then we have that $d_{in}^A(u) \leq |C_A(u)|/3$, so $d_{in}^A(u) \leq \frac{1}{2}\bar{d}_{in}^A(u)$, and hence $d(u) + d_{in}^A(u) = d_{out}^A(u) + 2d_{in}^A(u) \leq d_{out}^A(u) + \bar{d}_{in}^A(u)$. $\square$

**Lemma C.3.** *Consider a fixed ordering $\pi$ and a clustering algorithm $A$, and consider a $A$-bad cluster $C_A^\pi$. Then, its cost is at least $\frac{2}{3}(1-\gamma)|C_A^\pi|^2$.*

*Proof.* Recall that a bad cluster does not have any $(ref, \alpha)$-poor nodes. So, all the bad nodes in a bad cluster are either light or heavy. Let $C = C_{ref}^\pi$.

For a light node $u \in C$, the cost of $u$ is at least $2|C|/3$, since $d_C(u) \leq |C|/3$. For a heavy node $u \in C$, the cost of $u$ is at least $(\beta - 1)|C|/2$, since $u$ has at least $(\beta - 1)|C|$ neighbors outside $C$. So the cost of any bad node in $C$ is at least $\min\{(\beta - 1)/2, 2/3\} \cdot |C|$. There are at least $(1 - \gamma)|C|$ many bad nodes in $C$. Thus, the total cost of $C$ is at least $\min\{(\beta - 1)/2, 2/3\} \cdot (1 - \gamma) \cdot |C|^2$. Since $\beta \geq 4$, $\min\{(\beta - 1)/2, 2/3\} = 2/3$. $\square$

# D. Ommited proofs

## D.1. Proof of Lemma 3.4

It holds that $\Pr\left(\pi(v) \leq L'/d(v) \mid v \in P_{ref}^\pi\right) = 1 - \Pr\left(\pi(v) > L'/d(v) \mid v \in P_{ref}^\pi\right)$, and we upper-bound the latter probability:

$$\Pr\left(\pi(v) > L'/d(v) \mid v \in P_{ref}^\pi\right) = \frac{\Pr\left(\pi(v) > L'/d(v) \text{ and } v \in P_{ref}^\pi\right)}{\Pr\left(v \in P_{ref}^\pi\right)}$$

$$= d(v) \cdot \Pr\left(\pi(v) > L'/d(v) \text{ and } \forall u \in N(v) : \pi(u) > \pi(v)\right)$$

$$\leq d(v) \cdot \left(1 - \frac{L'}{d(v)}\right)^{d(v)+1}$$

$$\leq d(v) \cdot e^{-L'}.$$

In the derivation, we used the fact that the entries of $\pi$ are chosen independently and uniformly from range $[0, 1]$. Now note that $d(v)e^{-L'} \leq ne^{c \log n} < n^{-c+1}$, so Hence, $\Pr\left(\pi(v) \leq L'/d(v) \mid v \in P_{ref}^\pi\right) \geq 1 - n^{-(c-1)}$, as desired.

## D.2. Proof of Lemma 3.5

Let $L' = L/2\beta$. Let $A$ be the set of first $(1-\epsilon)C_{ref}(v)[good]$ good nodes in $C_{ref}(v)$ and let $B$ be the set of last $\epsilon C_{ref}(v)[good]$ good nodes in $C_{ref}(v)$. So $C_{ref}(v)[good] = A \cup B$.

First we introduce some notation: For two nodes $u$ and $w$ where $w$ is inserted before $u$, the degree of a vertex $w$ at the time where $u$ is inserted is denoted by $d^{(u)}(w)$. Recall that the current degree of $w$ is denoted by $d(w)$, and $d(w) \geq d^{(u)}(w)$.

Next, note that if the pivot $v$ comes after all the nodes in $A$, then since $v$ is a pivot in REFERENCE CLUSTERING by Lemma 3.4 we have that whp $\pi(v) \leq L'/d(v) \leq L/d(v)$ and so it scans its neighborhood and invokes EXPLORE$(v)$ which assigns all these nodes to $v$ as their pivot. So suppose that $v$ comes before the nodes in $B$.

Now consider a good node $u \in C_{ref}(v)[good]$ and suppose that $u$ comes after $v$ in the dynamic ordering. Since $u$ is not heavy, we have $d(u) \leq \beta|C_{ref}(v)| \leq \beta d(v)$.

We compute the probability that $u$ is assigned to $v$'s cluster and also that $u$ invokes EXPLORE$(v)$. In particular, we want to lower-bound the probability that $\pi(v) < \pi(u) \leq \frac{L}{d^{(u)}(u)}$ and $d^{(u)}(v) \leq \frac{L}{\pi(u)}$, conditioned on $\pi(v) < \pi(u)$, i.e., in REFERENCE CLUSTERING $u$ is in $C_v$.

First note that if $d(v) \leq L$ and $d(u) \leq L$, then both these conditions are satisfied. So we assume that $\max(d(u), d(v)) > L > L'\beta$.

Next, recall that since $v$ is a pivot in the REFERENCE CLUSTERING, by Lemma 3.4, we have that $\pi(v) \leq L'/d(v)$ holds with probability $1 - 1/n^{c-1}$ for a large constant $c$ by Lemma 3.4. Moreover, since $d^{(u)}(v) \leq d(v)$, the probability that $d^{(u)}(v) \leq \frac{L}{\pi(u)}$ is at least the probability that $d(v) \leq \frac{L}{\pi(u)}$. Similarly, the probability that $\pi(u) \leq \frac{L}{d^{(u)}(u)}$ is at least the probability that $\pi(u) \leq \frac{L}{d(u)}$ as $d^{(u)}(u) \leq d(u)$. So:

$$\Pr\left(u \text{ invokes EXPLORE}(v) \mid u \in C_{ref}(v)\right)$$

$$= \Pr\left(\pi(v) < \pi(u) \leq \frac{L}{d^{(u)}(u)} \text{ and } d^{(u)}(v) \leq \frac{L}{\pi(u)} \mid \pi(v) < \pi(u) \leq 1\right)$$

$$\geq \Pr\left(\pi(v) < \pi(u) \leq \frac{L}{d(u)} \text{ and } d(v) \leq \frac{L}{\pi(u)} \mid \pi(v) < \pi(u) \leq 1\right)$$

$$= \Pr\left(\pi(v) < \pi(u) \leq \frac{L}{\max(d(u), d(v))} \mid \pi(v) < \pi(u) \leq 1\right)$$

$$= \Pr\left(\pi(v) < \pi(u) \leq \frac{L}{\max(d(u), d(v))} \mid \pi(v) < \pi(u) \leq 1, \pi(v) \leq L'/d(v)\right) \cdot \Pr\left(\pi(v) \leq L'/d(v)\right)$$

$$+ \Pr\left(\pi(v) < \pi(u) \leq \frac{L}{\max(d(u), d(v))} \mid \pi(v) < \pi(u) \leq 1, \pi(v) > L'/d(v)\right) \cdot \Pr\left(\pi(v) > L'/d(v)\right)$$

$$\geq \Pr\left(\frac{L'}{d(v)} < \pi(u) \leq \frac{L}{\max(d(u), d(v))} \mid \frac{L'}{d(v)} < \pi(u) \leq 1\right) \cdot \left(1 - n^{-c+1}\right)$$

If $d(v) \leq d(u)$, then since $d(u) \leq \beta|C_{ref}(v)| \leq \beta d(v)$ and $L'\beta = L/2$, we have

$$\Pr\left(u \text{ invokes EXPLORE}(v) \mid u \in C_{ref}(v)\right) \geq \Pr\left(\frac{L'\beta}{d(u)} < \pi(u) \leq \frac{L}{d(u)} \mid \frac{L'\beta}{d(u)} < \pi(u) \leq 1\right) \cdot \left(1 - n^{-c+1}\right)$$

$$= \frac{L - L'\beta}{d(u) - L'\beta} \cdot \left(1 - n^{-c+1}\right) \geq \frac{L - L'\beta}{d(u)} \cdot \left(1 - n^{-c+1}\right)$$

$$\geq \frac{L/2}{d(u)} \cdot \left(1 - n^{-c+1}\right) \geq \frac{\alpha L/2}{d(u)} \cdot \left(1 - n^{-c+1}\right)$$

Nnote that $d(u) - L'\beta > 0$ since we assume that $\max(d(u), d(v)) > L > L'\beta$.

If $d(v) \geq d(u)$, we use the fact that since $u$ is not $(\alpha, A)$-poor, we have $d(u) \geq \alpha d(v)$, and so

$$\Pr\left(u \text{ invokes EXPLORE}(v) \mid u \in C_{ref}(v)\right) \geq \Pr\left(\frac{L'}{d(v)} < \pi(u) \leq \frac{L}{d(v)} \mid \frac{L'}{d(v)} < \pi(u) \leq 1\right) \cdot \left(1 - n^{-c+1}\right)$$

$$\geq \frac{L - L'}{d(v) - L'} \cdot \left(1 - n^{-c+1}\right) \geq \frac{L/2}{d(v)} \cdot \left(1 - n^{-c+1}\right)$$

$$\geq \frac{\alpha L/2}{d(u)} \cdot \left(1 - n^{-c+1}\right)$$

The above inequality holds for each $u \in B$. We show that with high probability, a node in $B$ invokes EXPLORE($v$). First recall that $C_{ref}(v)$ is a good cluster, so $|C_{ref}(v)[good]| \geq \gamma|C| \geq \gamma d(u)/\beta$. Thus we have $|B| \geq \gamma \epsilon d(u)/\beta$. Let $t = \frac{\epsilon \gamma \alpha}{4\beta}$. Since $(1 - n^{-c+1}) > 1/2$, we have:

$$\Pr\left(u \text{ invokes EXPLORE}(v)\right) \geq \frac{tL}{|B|}$$

So the probability that none of $u \in B$ invokes EXPLORE($v$) is at most $(1 - \frac{tL}{|B|})^{|B|} \leq e^{-tL} \leq n^{-c}$, where we use $L \geq \frac{4c\beta}{\epsilon \gamma \alpha} \log n = \frac{c}{t} \log n$. So with probability $1 - n^{-c}$, EXPLORE($v$) is invoked by a node in $B$, and so all the nodes in $A$ are going to be assigned to $v$ as their pivot.

### D.3. Proof of Lemma 3.7

Let $k = C^*$ and let $t$ be the smallest power of $(1 + \epsilon)$ no smaller than $\frac{2k}{3}$. So $\frac{2k}{3} \leq t \leq \frac{2k(1+\epsilon)}{3}$. Let $S = C^* \setminus C_t$, and let $T = C_t \setminus C^*$. We refer to the clustering where $C_t$ is one cluster and all $B_v \setminus C_t$ is singleton as $\mathcal{C}_1$ and the clustering where $C^*$ is one cluster and all $B_v \setminus C^*$ is singleton as $\mathcal{C}^*$.

The cost of $\mathcal{C}_1$ and $\mathcal{C}^*$ differ in edges and non-edges with one endpoint in $S$ or $T$. They share any other cost associated to an edge or non-edge that is in disagreement with the clustering. So we only need to compare the excess cost that is not part of this common cost.

By the cost of a node $u \in S \cup T$, we mean the number of $v \notin S \cup T$ where $uv$ is in disagreement with the clustering plus half the number of $v \in S \cup T$ where $uv$ is in disagreement with the clustering. We let $cost(u)_{\mathcal{C}}$ denote the cost of $u$ in a clustering $\mathcal{C} \in \{\mathcal{C}_1, \mathcal{C}^*\}$. Let $cost(\mathcal{C})$. Note that the total cost of $\mathcal{C}$ is the sum of the individual costs of $u \in S \cup T$, plus the common cost between $\mathcal{C}_1$ and $\mathcal{C}^*$.

First, assume that $|T| \leq k$. We show that $cost(\mathcal{C}_1) \leq \frac{4}{1-2\epsilon} cost(\mathcal{C}^*)$.

Consider a node $u \in S$. In $\mathcal{C}_1$, the cost of $u$ is at most $t$ since each node in $S$ has degree at most $t$, and $u$ is a singleton in $\mathcal{C}_1$. In $\mathcal{C}_2$, the cost of $u$ is at least $\frac{k-t}{2}$ since there are at least $k - t$ non-edges attached to $u$ in $C^*$. Note that for $t \leq \frac{2k(1+\epsilon)}{3}$, we have $cost(u)_{\mathcal{C}_1} \leq t \leq \frac{4}{1-2\epsilon} \cdot (k - t)/2 \leq \frac{4}{1-2\epsilon} cost(u)_{\mathcal{C}_2}$.

To compute the cost of $u \in T$, for a set $C$ recall that $d_C(u)$ is the degree of $u$ inside $C$. So $d(u) - d_{C_t}(u)$ is the number of edges attached to $u$ outside $C_t$. In $\mathcal{C}_1$ the cost of $u$ is the number of nodes attached to $u$ outside of $C_t$ plus the number of nodes $v$ not attached to $u$ inside $C_t \setminus T$, plus half of the number of nodes $v \in T$ that are not attached to $u$. So

$$cost(u)_{\mathcal{C}_1} = d(u) - d_{C_t}(u) + (|C_t| - |T| - d_{C_t \setminus T}) + (|T| - 1 - d_T(u))/2$$

$$\leq d(u) - d_{C_t}(u) + (|C_t| - |T| - d_{C_t \setminus T}) + (|T| - 1 - d_T(u))$$

$$= |C_t| + d(u) - 1 - 2d_{C_t}(u)$$

$$\leq |T| + |C^* \cap C_t| + d(u) - 2d_T(u)$$

$$\leq 2k + t - 2d_T(u)$$

Meanwhile we have that there are at least $d(u) - d_T(u)$ nodes attached to $u$ outside of $T$, so we have

$$cost(u)_{\mathcal{C}_2} = d(u) - d_T(u) + d_T(u)/2 \geq \frac{t - d_T(u)}{2}$$

Note that when for $t \geq 2k/3$, we have $2k + t - 2d_T(u) \leq 4(\frac{t - d_T(u)}{2})$, so $cost(u)_{\mathcal{C}_1} \leq 4 cost(u)_{\mathcal{C}_2}$. And so $cost(\mathcal{C}_1) \leq \frac{4}{1-2\epsilon} cost(\mathcal{C}^*)$.

Now suppose that $|T| > k$. Let cluster $\mathcal{C}_2$ be the clustering where all the nodes in $B_v$ are singleton. In fact, $\mathcal{C}_2$ is the clustering for when $t = n$. In this case, the cost of $\mathcal{C}_2$ is the cost of $\mathcal{C}^*$ plus the number of edges between the nodes in $C^*$. We have $cost(\mathcal{C}_2) \leq cost(\mathcal{C}^*) + k^2/2$. Now to bound $cost(\mathcal{C}^*)$, the cost of $cost(\mathcal{C}^*)$ is at least the number of edges with at least one endpoint in $T$. This value is at least $t|T|/2$. So $cost(\mathcal{C}^*) \geq t|T|/2$. Now since $|T| \geq k$ and $t \geq 2k/3$, we have $3 \cdot t|T|/2 \geq k^2/2$, so $cost(\mathcal{C}_2) \leq cost(\mathcal{C}^*) + k^2/2 \leq cost(\mathcal{C}^*) + 3 \cdot t|T|/2 \leq 4cost(\mathcal{C}^*)$.

## E. Running Time Analysis

In Appendix B, we show that our RECOMPUTE subroutine takes $\text{poly}(\log n, 1/\varepsilon)$ amortized time per update. Next, we analyze the running time of Algorithm 3 and prove Theorem 2.2.

*Proof.* We first introduce some notation: Let the degree of a node $w$ at the time of the insertion of node $u$ be $d^{(u)}(w)$. Recall that $d(w)$ is the current degree of $w$ and $d(w) \geq d^{(u)}(w)$.

First note that if $\pi(u) > L/d^{(u)}(u)$, then the running time is $O(\log n)$ plus the running time of UPDATE-CLUSTER, which is in $\text{poly}(\log n, \frac{1}{\varepsilon}) \leq \text{poly}(\log T, \frac{1}{\varepsilon})$ time; see Appendix A.2.3 for details.

Next, consider the case where $\pi(u) \leq L/d^{(u)}(u)$. For each such $u$, we set aside a budget of $L/\pi(u) \cdot \text{poly}(\log n, 1/\varepsilon)$. In this case, the algorithm scans all the neighbors of $u$, which takes $d^{(u)}(u) < L/\pi(u)$ time.

Next, we analyze the running time of EXPLORE. For this, consider a pivot node $v$. Note that when we run EXPLORE on a node $v$, we scan all its neighbors, and if a neighbor $w$ is also a pivot, we scan all of the neighbors of $w$ as well and remove $w$ from being a pivot. Observe that once a node is removed from being a pivot, it can never be a pivot until the next recompute, when nodes get new ranks. We pay for scanning the neighbors of $w$ from the budget of $w$, not $v$. This way, when a node $v$ runs EXPLORE, it only needs to pay at most $d(v)$ from its budget.

Now note that a node $v$ might run EXPLORE multiple times. In particular, $v$ runs EXPLORE either when we are processing $v$, or when a neighbor of $v$, say $u$ is being processed, and $d^{(u)}(v) < L/\pi(u)$. In the first case, we pay the cost of EXPLORE from $v$'s budget. In the second case, we pay the cost from $u$'s budget. Note that for a pivot node $v$, not only we have $d^{(v)}(v) < L/\pi(v)$, but also we have $d(v) < L/\pi(v)$ (see Lemma 3.4) whp.

So, in total, a pivot node $v$ only spends a budget of at most $d(v) \cdot \text{poly}(\log n, 1/\varepsilon) < L/\pi(v) \cdot \text{poly}(\log n, 1/\varepsilon)$: it spends $d^{(v)}(v) \leq d(v)$ for scanning all its neighbors when it is being inserted, $d^{(v)}(v) \leq d(v)$ when it runs EXPLORE, at most $d(v)$ (possibly) when another pivot $w$ runs EXPLORE and removes $v$ from being a pivot, and $d^{(v)}(v) \text{poly}(\log n, 1/\varepsilon) \leq d(v) \text{poly}(\log n, 1/\varepsilon)$ for running BREAK-CLUSTER (Theorem A.6).

A non-pivot node $u$ spends at most $2L/\pi(u)$: once for scanning all its neighbors and once for paying for EXPLORE for its pivot $v$.

Since $L = O(\log n)$, each node $u$ spends $\text{poly}(\log n, 1/\varepsilon) + O(\log n/\pi(u))$. Given that $\pi(u)$ is chosen uniformly at random from the range $[0, 1]$, in expectation, a node spends $\text{poly}(\log n, 1/\varepsilon) \leq \text{poly}(\log T, 1/\varepsilon)$ running time. $\square$

## F. Experiments-continued

### F.1. Approximation guarantee results

Below we include the approximation guarantee on the three remaining SNAP graphs.

### F.2. Running Time

Even though theoretically, our running time (and DYNAMIC-AGREEMENT running time) is faster than REFERENCE CLUSTERING, this advantage only appears in very big graphs. This is because both algorithms have a lot of bookkeeping, which means that the constant behind the $O(\log n)$ running time is rather big. Nevertheless, we show in Table 2 and Table 3 that SPARSE-PIVOT is faster than DYNAMIC-AGREEMENT.

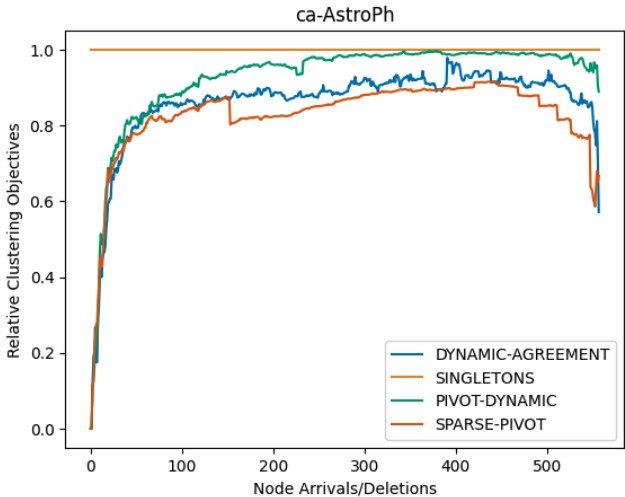

*Figure 2.* Correlation clustering cost for ca-astroph graph

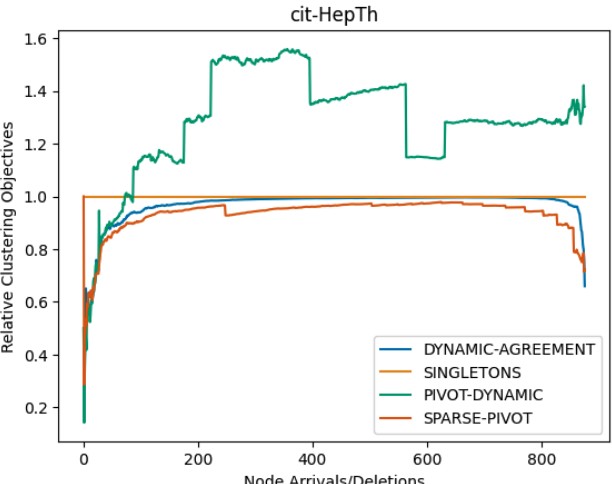

*Figure 3.* Correlation clustering cost for cit-hepth graph

| Density | DA | SP |
|---------|-------|-------|
| 253.36 | 36.75 | 31.91 |
| 114.87 | 43.08 | 29.69 |
| 69.74 | 50.77 | 26.36 |
| 52.17 | 49.27 | 23.36 |
| 42.25 | 41.23 | 25.14 |

*Table 2.* Running time comparison on Drift dataset

| Graph | DA | SP |
|-------------|-------|-------|
| facecbook | 8.14 | 3.31 |
| email-Enron | 9.69 | 4.48 |
| cit-HepTh | 20.24 | 11.53 |
| ca-AstroPh | 15.15 | 3.84 |

*Table 3.* Running time comparison on SNAP datasets

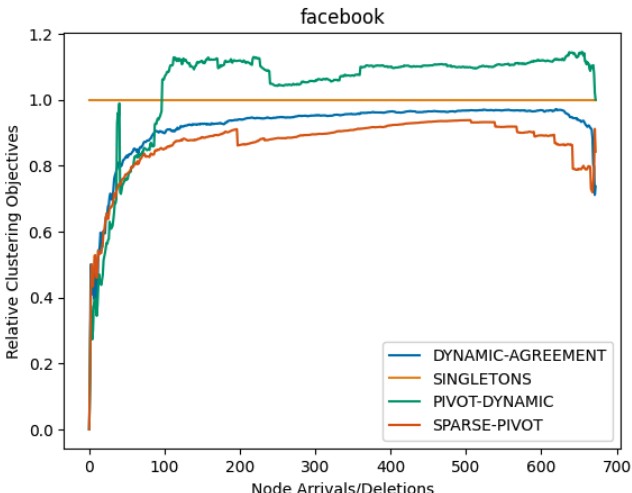

*Figure 4.* Correlation clustering cost for facebook graph

