# OpenReview forum: "Sparse-pivot: Dynamic correlation clustering for node insertions"
_ICML.cc/2025/Conference — ICML 2025 spotlightposter_

### Official Review · Reviewer_mgYi · 2025-03-13

**Overall Recommendation:** 4

**Summary:**

The authors consider the classic Correlation Clustering problem which, given a complete graph with edges labeled either + or -, the goal is to find a partition of the vertices so as to minimize the number of + edges across parts plus the number of - edges within parts.
The has received a lot of attention since its introduction in the early 2000s.

In this paper, the authors consider the dynamic setting where nodes are iteratively inserted (and not removed) into the dataset and the goal is to maintain a partition that minimizes the Correlation Clustering objective at all time, while minimizing the total running time (a.k.a update time).

The authors improve an ICML'24 paper which showed how to maintain a very large constant factor approximation to the problem while making polylog n database queries (here a database query is a query of one of the following types: (1) retrieving the degree of a node v; (2) selecting a random neighbor of v; and (3) checking whether two nodes u and v are connected by an edge). In this paper, the authors provide an algorithm with polylog n update time that achieves a 20+epsilon-approximation.

The algorithm is a refinement of the classic pivot algorithm for Correlation Clustering, called Sparse-Pivot, which combines ideas from previous work on streaming algorithms, post-processing the clusters obtained, and a new sampling strategy to define the pivots (looking at O(log n) random neighbors).

**Claims And Evidence:**

Yes

**Essential References Not Discussed:**

None

**Experimental Designs Or Analyses:**

Yes, comparison to previous work (ICML'24) is done carefully.

**Methods And Evaluation Criteria:**

Yes

**Other Comments Or Suggestions:**

No particular comments. Given that your algorithm seems to experimentally outperform ICML'24 and the pivot algorithm, I think it makes sense to experiment with it in a purely offline setting as well: could it be that your algorithm is the state-of-the-art for (offline) correlation clustering practical instances?

**Other Strengths And Weaknesses:**

The paper makes a significant improvement over the approximation ratio and interestingly this turns out to be pretty relevant for practice as well, despite previous pivot-based approaches being not so strong in practice (in fact, as shown by the authors, the vanilla pivot algorithm performs worst than all-singletons).
On the other hand, from a theoretical perspective, the technical novelty is not very high.

**Questions For Authors:**

I am wondering whether the result of the ICML'24 paper allows to define "admissible" edges in the sense of [1]. And so you could avoid the cleaning step by only performing pivot on the admissible edges. But that's maybe a long shot.


[1] Vincent Cohen-Addad, David Rasmussen Lolck, Marcin Pilipczuk, Mikkel Thorup, Shuyi Yan, Hanwen Zhang:
Combinatorial Correlation Clustering. STOC'24

**Relation To Broader Scientific Literature:**

Looks good to me.

**Theoretical Claims:**

Yes, I checked most of the proofs.

---

> ### Author Rebuttal · Authors · 2025-03-31
>
> > On the other hand, from a theoretical perspective, the technical novelty is not very high.
>
> We believe our algorithm is intuitive and relatively simple, which we view as a clear advantage—particularly from an implementation standpoint. However, proving that those natural ideas significantly improve approximation guarantees of previous work turned out to be challenging. We would be happy to see a simpler analysis, though, if there is one.
>
> > No particular comments. Given that your algorithm seems to experimentally outperform ICML'24 and the pivot algorithm, I think it makes sense to experiment with it in a purely offline setting as well: could it be that your algorithm is the state-of-the-art for (offline) correlation clustering practical instances?
>
> We do not think that our algorithm outperforms [1] in the offline setting.
> Intuitively, our algorithm outperforms Pivot because it “fixes” bad clusters made by Pivot. Those fixes are reflected by Break-Cluster method. Namely, if a cluster obtained by Pivot has many non-edges, Break-Cluster attempts to improve it by making some of the nodes singletons. We generally believe that studying this kind of local improvements is an excellent research direction. Some of the prior work, e.g., [1] that you referred to, performs this kind of local improvement, although not directly on Pivot clustering. We believe that there is room to perform relatively simple local improvements and also achieve an approximation better than 3.
>
> > I am wondering whether the result of the ICML'24 paper allows to define "admissible" edges in the sense of [1]. And so you could avoid the cleaning step by only performing pivot on the admissible edges. But that's maybe a long shot.
> [1] Vincent Cohen-Addad, David Rasmussen Lolck, Marcin Pilipczuk, Mikkel Thorup, Shuyi Yan, Hanwen Zhang: Combinatorial Correlation Clustering. STOC'24
>
> These are great questions and good directions for future work. Having admissible edges in the sense of [1] requires maintaining pre-clustering dynamically, which might be challenging.

---

### Official Review · Reviewer_aRAg · 2025-03-14

**Overall Recommendation:** 4

**Summary:**

This paper presents "SPARSE-PIVOT," a new dynamic correlation clustering algorithm designed for node insertions. The algorithm builds upon a variant of the PIVOT algorithm and aims to improve the update time and approximation factor compared to the existing state-of-the-art algorithm by Cohen-Addad et al. (ICML 2024). The main algorithmic idea is to combine a fast approximate pivot selection using random sampling with a refinement step that removes poorly clustered nodes. The paper claims an amortized update time of $O(\log^{O(1)}(n))$  and an approximation factor of $20 + \epsilon$. The theoretical analysis is complemented by experimental evaluation, showing better performance to previous methods.

**Claims And Evidence:**

The claims regarding the approximation factor and update time appear to be supported by the provided theoretical analysis. The evidence presented is convincing, with detailed proofs in the appendix. The experiment provides also a strong support, showing the benefits of the new method.

**Essential References Not Discussed:**

The paper appears to cite all essential references.

**Experimental Designs Or Analyses:**

I checked the soundness of the experimental setup. The choice of datasets, baselines, and metrics appear appropriate.

**Methods And Evaluation Criteria:**

The proposed methods and evaluation criteria make sense. The dynamic setting with node insertions is a relevant problem, and comparing against the algorithm by Cohen-Addad et al. (2024) is a reasonable benchmark. The use of both real-world and synthetic datasets is appreciated.

**Other Comments Or Suggestions:**

Overall, the strengths outweigh the weaknesses. The paper provides a valuable contribution to the field of dynamic correlation clustering with both theoretical and practical significance.

**Other Strengths And Weaknesses:**

**Strengths:**
  * The paper presents a novel and theoretically sound algorithm with a significantly improved approximation factor compared to the paper by Cohen-Addad et al.
  * Provides solid empirical evidence on real-world and synthetic datasets, showcasing the practical effectiveness and improved performance of SPARSE-PIVOT. In particular, the evaluation demonstrates that the algorithm performs better than the main competitor.
  *  I find the new adaptation of the pivot algorithm quite clever and insightful.
  *  The paper is well-written and clearly explains the algorithm, theoretical analysis, and experimental results. The appendices provide detailed support for the main claims.

**Weaknesses:** The database model seems a bit unrealistic, but to be fair this is a previously studied and accepted model.

**Questions For Authors:**

No questions.

**Relation To Broader Scientific Literature:**

The paper situates itself well within the correlation clustering literature. It builds upon prior work on PIVOT and addresses the dynamic setting, which has seen growing interest. In particular, the authors consider the dynamic database model introduced recently by Cohen-Addad et al (2024). The relationship to other dynamic algorithms is clearly acknowledged.

**Theoretical Claims:**

I partially checked the correctness of proofs in the supplementary section. I reviewed Lemma 3.2, 3.3, 3.4, 3.5. I did not spot any obvious flaw.

---

> ### Author Rebuttal · Authors · 2025-03-31
>
> We thank the reviewer for their kind and valuable feedback.

---

### Official Review · Reviewer_JpHJ · 2025-03-17

**Overall Recommendation:** 4

**Summary:**

This paper studies the correlation clustering problem on graphs in the dynamic setting. This variant of clustering is an important problem both in theory and practive, and has been extensively studied in different computational models.

Traditionally, most of the algorithms in the dynamic graphs literature consider the setting where edges of the underlying graphs undergo updates, i.e., they are inserted or deleted. This paper considers a more challenging and the less-studied setting of node insertions; when a node is added, then all its incident neighbours are revelaed at intsertion. Naturally, one could simple simluate node insertions by inserting all edges incident to the inserted node – but since nodes can have deegree up to O(n), this leads to undiserable times.

The paper under reivew studies algorithms that beat such trivial bounds. In fact, for the correlation clustering problem, they propose an algorithm that approximates the optimal correlation clustering objective up to a factor of ~20, and runs in poly-logarithmic amortized time per node insertions and even *random* deletions. The prioir work by Cohen-Addad et al. achieved similar runtime guarantees, but at the cost of a much constant approximation.

The starting point of their algorithm is a 5-approximate algorithm due to Behnezhad et al. (a variant of the well-known PIVOT algorithm) in the semi-streaming setting. The idea is to take this algorithm and adapt it to the dynamic setting. Some technical ideas involve: picking random ranks for each node, and using this rank to classify nodes depending on their degree, and finally doing some clean-up, so that bad nodes cannot belong to clusters, etc. While these algorithmic steps make sense for the problem at hand, the adaption and especially the analysis are quite cumbersome and require special care. These type of algorithms are in my opinon the *whole grail* in algorith mdesign; easy to describe and understand, but not so trivial to analyse!

I’m not very familiar with the algorithm by Cohen-Addad et al., but the algorirthm that the paper proposes looks very natural and *the right one* to me.

## update after rebuttal
Thanks for the rebuttal. I keep my score as is.

**Claims And Evidence:**

Everything looks fine.

**Essential References Not Discussed:**

I didn't come across any.

**Experimental Designs Or Analyses:**

everything is done properly; I also liked that the authors discussed some heuristic approaches to the theoretical algorithm that could lead to better peformance in practice. This work is an example of nice bridge between theory and practice;

One downside is that the emperical improvements over previous algorithms don’t seem that large in practice on the benchmark data-sets.

**Methods And Evaluation Criteria:**

Yes, the proposed methods and evaluation criteria make sense.

**Other Comments Or Suggestions:**

None.

**Other Strengths And Weaknesses:**

Strengths: core problem in dynamic clustering with applications in practice; the node insertion/deletion setting is more challenging than edge updates; the algorithm comes with strong guarantees and easy to understand

Weaknesses: perhaps matching the 5-approximation of PIVOT would have been more exciting, at least from a theory perspective.

**Questions For Authors:**

None.

**Relation To Broader Scientific Literature:**

Relevant to anyone working on clustering across all subcommunities of computer science.

**Theoretical Claims:**

I did a pass on some of the proofs. I didn't have time to thoroughly check the correctness -- the approach looks very feasible though.

---

> ### Author Rebuttal · Authors · 2025-03-31
>
> > One downside is that the empirical improvements over previous algorithms don’t seem that large in practice on the benchmark data-sets.
>
> On all the benchmark datasets, our algorithm always performs better than previous algorithms. In some cases, the running time improvement is 2-3x; please see Table 2 and Table 3 in the appendix.
>
> > Weaknesses: perhaps matching the 5-approximation of PIVOT would have been more exciting, at least from a theory perspective
>
> Indeed, this is quite an interesting direction for future work. The main bottleneck in achieving this with our approach and analysis is in Break-Cluster.

---

### Official Review · Reviewer_Tvij · 2025-03-17

**Overall Recommendation:** 4

**Summary:**

This paper introduces a new Correlation Clustering algorithm for node insertion / deletion in a dynamic setting. Upon node arrival its edges with all existing nodes are revealed, and we are allowed to make changes to the maintained clustering solution. The algorithm has constant approximation factor and sublinear amortized processing time per node arrival. which is a substantial improvement over the approximation factor of the algorithm by Cohen-Addad et al. Empirical results show the new algorithm to outperform state of the art benchmarks.

**Claims And Evidence:**

The claims are well supported.

**Essential References Not Discussed:**

There are other earlier papers that discuss online correlation clustering, although the settings are less general than this one, where changing previously made cluster assignment decisions are not allowed, but also studies pivot. Some of them are:

Mathieu, Claire, Ocan Sankur, and Warren Schudy. "Online correlation clustering." arXiv preprint arXiv:1001.0920 (2010).
Cohen-Addad, Vincent, et al. "Online and consistent correlation clustering." International Conference on Machine Learning. PMLR, 2022.
Lattanzi, Silvio, et al. "Robust online correlation clustering." Advances in Neural Information Processing Systems 34 (2021): 4688-4698.

**Experimental Designs Or Analyses:**

The experiment designs are quite simplified and could be improved in many ways. The results suffice to show that the proposed method outperforms state of the art, but it would be better if the authors also show how the running time scales with changing dataset sizes (number of nodes), the tradeoff between time and approximation ratio (controlled by $\epsilon$), etc.

**Methods And Evaluation Criteria:**

Yes.

**Other Comments Or Suggestions:**

The paper needs some proof-reading. There are some notations that appear without being mentioned. I couldn't find the definition of the BREAK-CLUSTER and UPDATE-CLUSTER. They have never been formally defined in the main body but directly used. The analysis can also be organized better. I would suggest putting two main theorems about the run-time and approximation guarantees at the beginning of the section so that the reader doesn't have to skim through the whole section to find what the guarantees are.

**Other Strengths And Weaknesses:**

The theoretical results are quite insightful for the online correlation clustering community. The algorithm design is also original and gives strong results.

**Questions For Authors:**

1. The algorithm assigns a uniformly random value between [0,1] to every node. The values are used to determine their relative order. Is this essentially the same with randomly shuffling the whole sequence of nodes offline, giving them ranks and treating the ranks as known information?
2. Do we have to know the size n in advance so that we can compute the threshold $L$ for the algorithm to work?

**Relation To Broader Scientific Literature:**

The paper mostly builds on existing Pivot-based algorithms for correlation clustering but innovates the algorithm. It can be seen as another variant of online correlation clustering. It is also broadly related to online clustering or general online algorithm problems.

**Theoretical Claims:**

Yes. I checked the major claims.

---

> ### Author Rebuttal · Authors · 2025-03-31
>
> > The experiment designs are quite simplified and could be improved in many ways. The results suffice to show that the proposed method outperforms state of the art, but it would be better if the authors also show how the running time scales with changing dataset sizes (number of nodes), the tradeoff between time and approximation ratio (controlled by ), etc.
>
> We agree that a comprehensive experiment set up would compare running time vs. dataset size. However, the focus of our paper is on theoretical results and validation of the approximation factor. We include experiments to have a full comparison with previous work.
>
> > The paper needs some proof-reading. There are some notations that appear without being mentioned. I couldn't find the definition of the BREAK-CLUSTER and UPDATE-CLUSTER. They have never been formally defined in the main body but directly used. The analysis can also be organized better. I would suggest putting two main theorems about the run-time and approximation guarantees at the beginning of the section so that the reader doesn't have to skim through the whole section to find what the guarantees are.
>
> Thanks for the suggestions on improving the writeup; we will incorporate them into the final version. On Page 3, we outline our main ideas. Idea number 3 is essentially Break-Cluster; we will emphasize this in the final version. It takes all the nodes assigned to a pivot v (set  B_v) and makes some of them singletons (so set C_v is the set of remaining nodes). Update-Cluster updates this set when a new node is assigned to pivot v. We will explain these subroutines before using them; right now, they are only described in detail in Appendix A.
>
> > The algorithm assigns a uniformly random value between [0,1] to every node. The values are used to determine their relative order. Is this essentially the same with randomly shuffling the whole sequence of nodes offline, giving them ranks and treating the ranks as known information?
>
> Yes, it is the same.
>
> > Do we have to know the size n in advance so that we can compute the threshold
> for the algorithm to work?
>
> That’s a great question. We do not need to know n in advance; that can be addressed by following standard techniques. Namely, when the graph size doubles, we recompute everything from scratch with the new n. This cost of recomputation is amortized over the run of the algorithm; specifically, it is charged to the steps when the current set of vertices arrived. We discuss a similar idea in Appendix B.

---

### Decision · Program_Chairs · 2025-05-01

**Decision:**

Accept (spotlight poster)

**Comment:**

This paper addresses dynamic correlation clustering with a significant improvement in approximation factors and sublinear update times. It builds upon Pivot-based methods, refining them with a novel procedure to handle poorly formed clusters. The algorithm achieves a constant-factor approximation and delivers superior empirical results, consistently outperforming prior approaches.

Reviewers commented on the clarity and organization, highlighting how it bridges theory and practice. Despite the algorithm’s seemingly simple design, the underlying analysis is nontrivial and yields substantial benefits in both solution quality and speed. Overall, this work marks an important advance in dynamic clustering research and suggests intriguing directions for future explorations.